# Complex network analysis to understand trading partnership in French swine production

**Pachka Hammami**[1], **Stefan Widgren**[2], **Vladimir Grosbois**[3,4,5], **Andrea Apolloni**[3,4,5], **Nicolas Rose**[1], **Mathieu Andraud**[1]*

**1** Anses Ploufragan-Plouzané-Niort Laboratory / Epidemiology, Health and Welfare Research Unit (EpiSaBE), French Agency for Food, Environmental and Occupational Health & Safety, Ploufragan, France, **2** Department of Disease Control and Epidemiology, National Veterinary Institute, Uppsala, Sweden, **3** Animal, Health, Territories, Risks, Ecosystems, Research Unit (ASTRE)/Agricultural Research for Development/Campus de Baillarguet, Cirad, Montpellier, France, **4** Animal, Health, Territories, Risks, Ecosystems, Research Unit (ASTRE), Univ. Montpellier, Montpellier, France, **5** Animal, Health, Territories, Risks, Ecosystems, Research Unit (ASTRE)/French National Institute for Agricultural Research/Campus de Baillarguet, INRAE, Montpellier, France

* mathieu.andraud@anses.fr

**Data Availability Statement:** Raw data are confidential and cannot be shared publicly. Data were collected through the National Swine Identification Database owned by BDporc. The

## Abstract

The circulation of livestock pathogens in the pig industry is strongly related to animal movements. Epidemiological models developed to understand the circulation of pathogens within the industry should include the probability of transmission via between-farm contacts. The pig industry presents a structured network in time and space, whose composition changes over time. Therefore, to improve the predictive capabilities of epidemiological models, it is important to identify the drivers of farmers' choices in terms of trade partnerships. Combining complex network analysis approaches and exponential random graph models, this study aims to analyze patterns of the swine industry network and identify key factors responsible for between-farm contacts at the French scale. The analysis confirms the topological stability of the network over time while highlighting the important roles of companies, types of farm, farm sizes, outdoor housing systems and batch-rearing systems. Both approaches revealed to be complementary and very effective to understand the drivers of the network. Results of this study are promising for future developments of epidemiological models for livestock diseases. This study is part of the One Health European Joint Programme: BIOPIGEE.

## Introduction

Livestock mobility is one of the important factors in the emergence and spread of infectious diseases [1–3]. Therefore, studying the structure of commercial exchanges between production holdings is particularly informative with regards to potential transmission routes [4]. Capturing interactions in heterogeneous populations is proving to be very effective in providing relevant information to develop network-based monitoring and control strategies [5, 6].

owner of the data can be contacted at: BDPorc, 43, rue Sedaine, CS 91115, 75538 PARIS cedex 11 – FRANCE (contact@bdporc.fr). Aggregated data were provided within the manuscript and its Supporting information files. The R package 'SwineNet' including code and example dataset is in final phase of development and will soon be available publicly on GitHub (https://github.com/Pachka/SwineNet).

**Funding:** This work was supported by funding from the European Union's Horizon 2020 Research and Innovation programme under grant agreement No 773830: One Health European Joint Programme. The funders had no role in study design, data collection and analysis, decision to publish, or preparation of the manuscript.

**Competing interests:** The authors have declared that no competing interests exist.

In France, the swine population is distributed among different facilities, including facilities of the industry, such as farms or slaughterhouses, but also private or event-related facilities, such as zoos. All these facilities are connected by the trade of live animals, thus forming a complex network whose drivers and structure can be analyzed. In the swine industry, the timing of animal movements follows strict rules, with a specific residence time in each production stage defined by the farming system. However, while the movement schedule is deterministic, the destination of animals is left to the farmer's choice. Even if, largely established by contractual relationships, the global contact patterns do not vary to a large extent, the loyalty of agricultural trading partners can be somehow volatile [7, 8]. Therefore, even if the global topology of the network is stable over time, the between-farm movements change, owing to specific drivers such as holding types [9]. The network observed at time $t$ will have the same topology/structure as the network observed at time $t + 1$ but could be composed of a different set of between-farm movements; for instance, movements observed in 2019 will probably differ from those observed in 2015, but the global structure and properties of both networks should remain similar. Therefore, an in-depth study of the network of farm-to-farm contacts would provide a better understanding of the spread of infectious livestock diseases. Identifying the key factors that determine farmers' choices of trade partnerships could provide relevant information for simulating realistic connections within the production chain and identify potential hot spots for prevention, surveillance and control.

Since the mid-1980s, during the first HIV epidemics [10], network analysis methods have been widely used to study the spread of infectious diseases and has become more and more common over the years [2, 4, 8, 11–14]. They have proven to be powerful to describe static and, recently, dynamic networks [15–18]. Different methods have been developed to predict links in complex networks [19, 20]. These methods usually take into account either structural metrics/attributes of the networks or characteristics of nodes and edges, but rarely all at the same time. Network drivers, affecting the contact probabilities between farms, can be analyzed using more advanced tools such as exponential random graph models (ERGMs) that have been under development in recent years [21–23]. ERGMs are a class of statistical models based on exponential-family graph theory aiming to analyze network data. By specifying the probability distribution for a set of random networks, it allows to highlight the most probable contacts in a structured population. They appear to be appropriate tools for analyzing farm interactions by taking into account both network topology, and farms and movements' characteristics [24, 25].

This study uses complex network analysis (CNA) methods, including the fitting of ERGMs, to describe French pig farm network's characteristics and determine drivers of trade partnerships.

## Materials and methods

### Data

**Database description.** As a part of their activities of disease surveillance and control, the French Ministry of Agriculture has established a database on Swine owners and movements since 2009 (BDporc). This database is managed by different professionals in the pig industry and contains information about the pigs holdings in France, as well as livestock movements among them. For this study we used i) the 2012–2014 data pre-processed for a previous analysis [8], this dataset includes farms, slaughterhouses, gathering centers, traders, markets, stopping points and agricultural cooperatives (companies), ii) annual farm exports recorded from 2015 to 2020, iii) slaughterhouses, rendering plants and gathering centers information exported from the database in March 2021.

**Holdings**. The details of the holdings were based on different exports from the database. In the databases, swine holdings are reported with their own coordinates. Two holdings more than 500 meters apart were considered as different holdings, even if they belonged to the same farm/owner. The datasets include a set of elements characterizing the agricultural activities of each establishment. These characteristics include: i) the owner identification, ii) the holding identification, iii) the geographical information including address, post code/city, country and GPS coordinates, iv) the production type, v) the main husbandry activity, vi) the reared species, vii) the eventual outdoor areas, viii) the respective sizes of reproduction, post weaning and fattening areas and ix) the eventual company identification (see details in Table 1). Slaughterhouses, rendering plants and gathering centers information contained identification number, address, city and owner.

**Movements**. Movement information is continuously recorded by transporters, farmers and administrations: every time an animal is moved from one holding to another the transport's ID number, places of origin, staging and destination, number and type of animals transported (piglet, pig, barrow, sow or cull), and the date are recorded. This information can be aggregated to establish the flows of animals between different holdings. Used datasets were extracted from the annual pig movements database from 2017 to 2020.

**Data cleaning and pre-processing.** The different datasets recording holdings information were cross-referenced by saving the most recent record when an holding appeared in several databases. Geographical coordinates were checked and, when missing, town center coordinates were used using OpenStreetMap Nominatim via the R package tmaptools [26]. Holdings without coordinates or town information were removed. The type of some farms was adapted according to the size of the reported areas (e.g. a finisher reporting post-weaning animals was changed to post-weaner/finisher). The status of multiplier and nucleus was used to define the type of farm independently of the type of farm initially declared.

The movement database included both national and international movements. However, exports and imports of foreign animals were recorded at country level without specific geographical coordinates. Therefore, movements from/to foreign countries were only used for descriptive purposes and were removed from further analysis. Movements for which essential information was missing (e.g. transport identification numbers, holdings identification numbers or animal category) were also removed from the analysis. Movements related to direct sales (from the producer to the consumer without going through commercial channels), domestic slaughter and unmarked animals were also removed. Assuming that movements

**Table 1. Details of holdings characteristics.**

| Characteristics | Options |
|---|---|
| Production types | Production for sale, Multipliers, Nucleus, Production for personal consumption, Production neither for sell nor consumption, Boar station, Wild boars. |
| Husbandry activities | Fattening, Post-weaning-Fattening, Farrowing-Fattening, Farrowing, Leisure pigs (pet), Post-weaning, Farrowing-post-weaning, Wild boar production, Laboratory/Research, Show pigs (park, refuge, zoo. . .). |
| Reared species | Pigs or wild boars |
| Batch-rearing systems* (BRS) | 4-batchs, 5-batchs, 7-batchs, 10-batchs, 20-batchs |

* Also called batch management production systems, it is characterized by the number of batch reared simultaneously and the duration of weaning. It defines the within-farm dynamics and structure. As it was not available in the database, it was estimated from the import and export frequencies and the size of the holdings. This approximation was justified by the proportion of farms in each BRS in the total industry.

from/to the overseas territories might depend on a different agro-economic and social systems than on the mainland, they were also removed.

The movement and holdings databases were cross-referenced. Holdings involved in at least one loading or unloading event during the period under study (2017–2019) were considered active. Only active holdings and movements between them were kept in the analysis. By looking for consistency between the types of animals shipped and the types of farms, inconsistent movements were removed (e.g. movements of barrows from farrowers were removed as well as movements of piglets from finishers). Finally, as the movement database is composed of ordered loading and unloading events, only movements with at least one loading event and one unloading event starting with a loading were kept.

To simplify the analysis, the transit of trucks without any loading or unloading event was not taken into account, thus focusing on diseases transmitted by direct contact. For each transportation, the records were divided according to the types of animals transported. For each type of animal, directed movements were created from the loading farms to the unloading farms when the loading event took place before the unloading events, as described in the method for the animal introduction model [8].

## Network analysis

A general description of the movement patterns in the whole French swine population was performed, whilst the in-depth analysis was limited only to the industrial sector. The swine industry is made up of different actors, including farms, slaughterhouses and traders. Highly structured in space and time, it is composed of two main sectors: the breeding sector including boar stations (insemination centre) and nucleus that supply gilts, nulliparous female pigs and boars to the multipliers, which produce pigs and breeding sows to supply the growing sector. The growing sector is composed of several types of farms ensuring the different production stages from farrowing to fattening. Depending on their type and size, each farm may consist of a gestation/breeding area, a maternity/farrowing area, a growing/weaning area and/or a finishing/fattening area. Animals are moved over time from one production area to another, either within the same farm or from one farm to another, before being sent to slaughter.

For the network analysis, the pig industry was represented as a set of nodes/holdings defined as geographical places where live animals are gathered. Those nodes are connected by edges/movements of live animals due to trade. The complex network formed by the interconnected pig holdings is characterized by endogenous (structural characteristics) and exogenous (node and edge characteristics) processes. In order to understand the main drivers of between-farm movements/farm connectivity—in other words, what motivates farmers' import and export choices, and to highlight the central industry actors that should be targeted for surveillance and control, a comprehensive descriptive analysis of the interactions was carried out, followed by a more advanced analysis using exponential random graph models.

**Exploratory analysis.**   Animal movements were analysed using complex network analysis (CNA) tools. The analysis was performed using R software [27] and more specifically the igraph package [28]. Several directed and unweighted networks were built using specific subsets of movements depending on the objectives.

The general description of the network was carried out for the whole period 2017–2020. To serve epidemiological concerns, the CNA was then focused on industrial pig production only including commercial farms and slaughterhouses. The network indicators, described in Table 2, were calculated to describe the general structure of the network in relation to our epidemiological concerns. The industry being composed of disconnected clusters (connected components), the closeness was not considered. In order to preserve the complexity of the

**Table 2. Networks metrics calculated for each network to analyse network structure.**

| Indicator | Description |
| --- | --- |
| Vertices number | Number of active holdings in the network |
| Edges number | Number of movements in the network |
| Median indegree | Median number of incoming movements per holding. The median was calculated only considering holdings receiving animals (indegree $\neq$ 0) |
| Median outdegree | Median number of outcoming movements per holding. The median was calculated only considering holdings exporting animals (outdegree $\neq$ 0) |
| Average relative betweenness | Average frequency with which a holding falls between pairs of other holdings on the geodesic (shortest) path connecting them divided by the maximum value it can take in the network |
| Weakly connected components | Groups of holdings where every pair of nodes is connected by a directed path. |
| Strongly connected components | Groups of holdings where every pair of nodes are connected either by a directed path or by a succession of directed path. |
| Transitivity | Probability that the adjacent holdings of a holding are connected to each other. Transitivity is only defined for undirected networks, therefore, we considered that all triadic configurations were transitive, the link direction was ignored. |
| Density | Ratio of the number of recorded movements and the total number of possible movements for active holdings |
| Diameter | Largest geodesic distance in the network—maximum shortest path length between two holdings |
| Degree assortativity | Pearson correlation coefficient between the degrees of linked holdings |
| Reciprocity ratio | Proportion of mutual connections in the network |
| Average path length | Average number of movements along the shortest paths (or geodesics) between all pairs of nodes |
| Proportion of nodes with the average Jaccard similarity coefficient equal to zero | Average number of common neighbours of two holdings divided by the number of neighbours of each of the two considered holdings. A neighbour being an adjacent holding |
| Average distance | Average Euclidian (geographical) distance per pairs of holdings |

network and to extract meaningful information, the stability of the network structure over time was assessed. For this purpose, and considering the fact that trade in the pig production sector in European countries is generally not seasonal [29–31], network indicators were calculated for various time windows and compared: annual networks, half-yearly networks and monthly networks.

The observed network during the second half of 2019 was then analyzed at three different scales: i) as a whole, and then divided into sub-networks to better analyze inter-farms interactions: ii) based on both production sectors: a breeding sub-network limited to nucleus, multipliers and boar stations, and a growing sub-network limited to farms raising pigs to sell them for consumption, and iii) based on the type of transported animals: sows (including breeding sows, gilts and nulliparous female pigs), piglets (defined as 8kg animals) and growing pigs (defined as 25kg animals). Between-farm movements drawn by barrow and culled animals were not considered because of the uncertainty on their consistency. The boar stations, producing semen for artificial insemination, were considered as epidemiological dead ends and removed from the analysis. Indeed, their role in animal movement is negligible. Furthermore, these nodes are submitted to strict biosecurity protocols and frequent tests to avoid pathogen transmission in semen. For all these reasons, owing to our objectives to derive the key features

of pig movement network in view to analyse the impact on pathogen spread, boar stations were effectively ignored in the analysis.

**Random networks simulation using exponential random graph model.** An exponential random graph model was selected and fitted for each sub-network based on the type of transported animals. ERGM is a class of statistical model for network analysis and simulation. It aims to analyse the effect of the network topology and the characteristics of its nodes and/or links on the presence (and absence) of network ties (movements in our case) between 2 nodes (pig holdings in our case). Based on those features, ERGMs models the network, identifying the probability distribution of any network so that large samples of networks can be generated.

**Model selection**. The analysis was performed using the R software [27] and more specifically the statnet and ERGM packages [32, 33]. A set of explaining variables was selected for each ERGM by sequential tests, following a bidirectional stepwise procedure. Starting with a forward approach from the simplest model including only edges as a predictor, the addition of each network statistic was tested sequentially, to identify a relevant subset of predictors (network statistics) for each final model. Then, a backward approach was used sequentially removing the previously selected predictors from the model to keep only those improving the model fit. Forward and backward approaches were implemented in turn until the Akaike Information Criterion (AIC) indicator of the model parsimony could not be improved by adding or removing any variable. Model fits were only considered when the ERGM fit stopped after the first estimation (using the maximum pseudo-likelihood estimates, MPLEs) or when the maximum likelihood estimation (MLE) converged without generating any warning (using a Markov Chain Monte–Carlo estimation algorithm (MCMC) with a Metropolis-Hastings sampler). To face computational issues related to the fitting of ERGMs on big networks, most of selected attributes were dyad-independent (with only covariate effects) estimated through MPLE on the complete network. Those estimation induced faster estimations than the Monte Carlo maximum likelihood estimation (MCMLE) [34].

**Goodness of fit, cross-validation and visual comparison**. In-sample performance was assessed for each selected ERGM by a Goodness of fit (Gof) analysis comparing six structural statistics of the observed network to those of networks randomly simulated by the fitted ERGM [35, 36]. 4000, 5000 and 6000 networks were randomly simulated to achieve the Gof for sows, piglets and growing pigs sub-network models respectively. The number of simulated networks were adjusted to stabilize the output, it depended on the size of the sub-network. The compared statistics were the in-degree, the out-degree, the minimum geodesic distance, the triad census distribution, the edge-wise shared partners and the dyad-wise shared partners. Those statistics are commonly used to test the Gof of ERGMs fitted to directed networks. They reflect the clustering of the network that can impact the disease spread dynamics [36]. Random simulated networks were also mapped and visually compared to sub-networks observed during the second semester of 2019 (6-months period from July to December). Finally, out-of-sample performance was assessed for each selected ERGM by a cross-validation approach comparing simulated statistics and maps to those of the networks observed on other semester/6-months period (before and after 2019).

## Results

### Data

Information were gathered on a total of 28,061 pig holdings and 1,426,865 transportation composed of 3,606,086 loading and unloading records occurring between 01/01/2017 and 31/12/2020. In total, 372,608 loading/unloading records (10%) were removed from the initial records for one or several of the following reasons: related to international transportation (25,073

records—1%), related to direct selling, household slaughter, unmarked animals (14,982 records—0%), missing site or transport identification (15,643 records—0%), related to an unrecorded holding (63,256 records—2%), related to removal of animals (death or missing) (148,063 records—4%), missing loading or unloading record (108,668 records—3%), related to one of the 303 holdings belonging to overseas territories (29,223 records—1%). In total, 10,442 holdings (37% of the initial dataset) were not involved in any movement record either because they did not report their movements or because they stopped raising pigs before 2017. Those holdings were considered as inactive and were removed from the analysis. The final dataset contained 3,234,289 loading and unloading records involving 17,316 holdings.

### Network analysis

**General desciption.** Most of the holdings involved in national movements of pigs were farms raising pigs for consumption, either to sell (86%) or for personal consumption (4%). Almost 94% of the active holdings were part of the pig industry, including multipliers (1.37%), nucleus (0.57%), boar stations (0.33%), farrowers (2.55%), farrowers-post-weaners (2.09%), farrowers-to-finishers (34.20%), post-weaners (0.56%), post-weaners-to-finishers (14.58%) and finishers (37.36%). Most of the active farms were 'finisher', 'farrowers-to-finishers' or 'post-weaners-to-finishers' (Fig 1 on the left). International movements involved 42 countries. Most of the international trade involved European countries, with international exports heading to Belgium (56%), Germany (15%), Italy (10%), Spain (8%) and the Luxembourg (5%) respectively, and international imports coming from Belgium (57%), Denmark (17%), the Netherlands (12%), Germany (7%) and Spain (5%) respectively.

Focusing on the pig industry (boar station: BS, nucleus: NU, multipliers: MU, farrowers: FA, farrowers-to-finishers: FF, finishers: FI, farrowers-post-weaners: FPW, post-weaners: PW, post-weaners-finishers: PWF), the total number of active farms varied on average by 1.7% per year. For more information, see S1–S3 Tables. Most of the movements (Fig 1 on the right) were headed to slaughterhouses (75.8%), followed by farms (19%) and gathering centers (4%). Sows were the main animal category involved in between-farm movements (35% of movements), followed by piglets (32%) and growing pigs (30%). The main observed patterns were the export of barrows from FF (34.26%), FI (15.24%) and PWF (13.73%) to slaughterhouses. The median number of transported animals varied with the type of transported animals as well as the origin and the destination sites. The median and 95% confidence intervals for the numbers of piglets, growing pigs, barrows and sows unloaded into a farm were respectively 172 [30; 678], 180 [8; 594], 14 [1; 257] and 8 [1; 40], whereas batches of 15 [1; 114], 3 [1; 126], 100 [1; 211] and 2 [1; 72] pigs were unloaded at slaughterhouses.

The distribution of geographic distances between connected sites varied with the animal category. Focusing on between-farm movements, the median and 95% confidence interval for distance traveled by breeding pigs, piglets and growing pigs respectively were 99 km [7; 545], 55 km [3; 236] and 39 [1; 249]. We can also note that the distance of exchanges between breeders (MU, NU, BS) was generally higher (154 km [2; 590]) than between growers (FA, PW, FI, FPW, PWF, FF: 43 [1; 261]).

**Analysis of observed networks.** The distribution of active holdings and movements was not homogeneous in space, e.g. most of the pig industry was gathered in north-western France (Fig 2). Monthly, semestrial and annual network metrics were calculated for the whole industry and the five sub-networks. For more information, see S1–S18 Tables.

The global number of active holdings (including all facilities, not only farms) per year increased from 2017 to 2019 (1.12% in 2018 and 0.82% in 2019) and then decreased in 2020 (-3.64%). The same tendency was observed in the industry with active farms annually varying

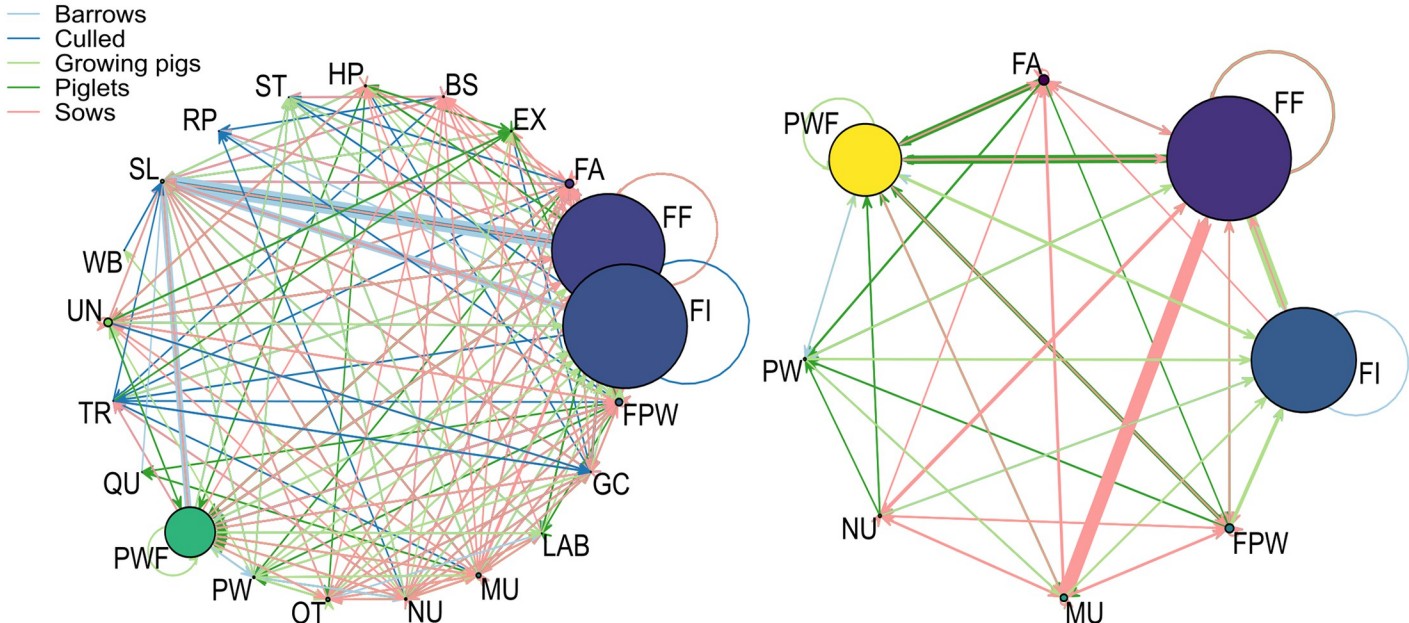

**Fig 1. Aggregated and directed French swine networks.** Number of animals exchanged between the different actors of the swine production by type of animal: total network (left), between-farm network (right). Colors of the arrows correspond to different type of animal, thickness of the arrows corresponds to the amount of animals. Colours of the circles correspond to different type of holdings and their size corresponds to the relative number of farms of that type. BS: boar station, EX: exposition (zoo, ...), FA: farrowers, FF: farrowers-to-finishers, FI: finishers, FPW: farrowers-post-weaners, GC: gathering center, HP: hobby pigs, LAB: laboratory, MU: multipliers, NU: nucleus, OT: other, PW: post-weaners, PWF: post-weaners-finishers, QU: Quarantine center, RP: rendering plant, SL: slaughterhouse, ST: Stopping point, TR: trader, UN: unknown, WB: wild boar.

by 1.33%, 0.68% and -3.52%. Conversely, the number of movements, as well as the number of loaded animals, only decreased during the studied period (Table 3).

**Complex networks analysis.** Focusing on understanding farmers' choices of trading partners, variables were limited to node and link attributes, with the exception of edges used to stabilize the number of links generated. Based on the analysis of the sub-networks observed in the second half of 2019, a set of variables (network statistic translated by an association of an ERGM term with a network attribute, see [21] for more information about terms) was identified as relevant to drive the network dynamics (Table 4). Network statistics included in the models were described in Tables 5 and 6. Model summaries including estimates and p-values, as well as model parameters are provided in supporting information, see S1–S3 Files.

**Whole network, breeding and growing sector sub-networks characteristics** (Fig 3). Most of the active holdings were part of the growing sector (FA, PW, FI, FPW, PWF and FF farms). The number of active holdings appeared to be stable over time regardless of the studied period. The ∼85,000 movements between farms per year mainly occurred within the growing sector (54,538 mov/year on average) and between the breeding (MU and NU farms and BS) and growing sectors, while movements within the breeding sector were in the minority (2,967 mov/year on average). Simplifying the networks by merging multiple movements did not change this pattern. There were fewer, smaller weakly- and strongly-connected components in the breeding sector than in the growing sector. While 99% of the connected components have less than 10 farms in the breeding sector, the growing sector is composed of connected components involving more than 150 farms. Although the median numbers of trading partners for either export (outdegree) or import (indegree) were similar in the breeding and growing networks, the median out-degrees in the growing network were slightly higher than in the

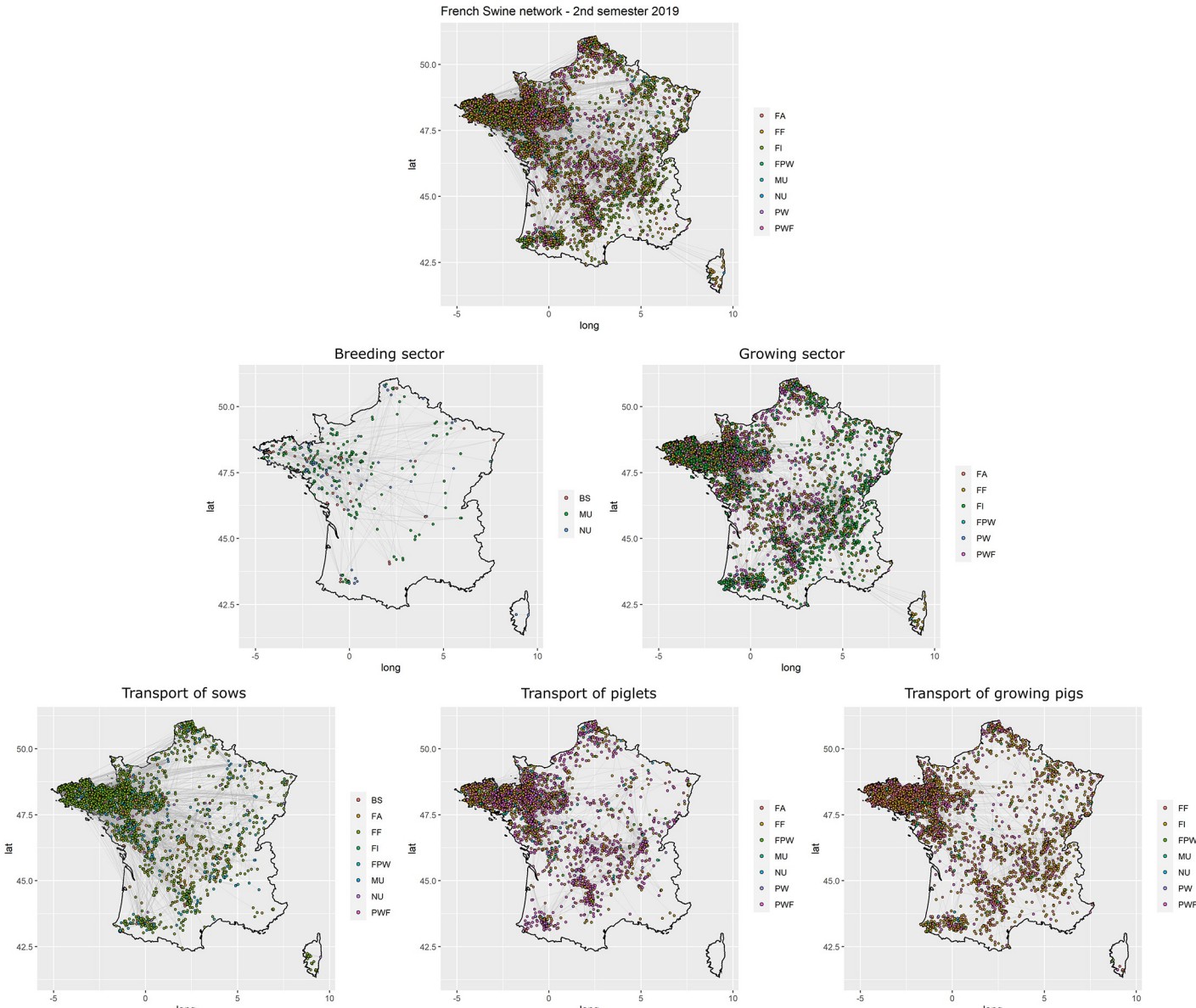

**Fig 2. Observed French swine networks during the second semester of 2019.** Mapping of active French swine holdings and animal movements during the second semester of 2019: whole industry (top), breeding sector (center-left), growing sector (center-right), transport of piglets (bottom-left), transport of growing pigs (bottom-center) and sows (bottom-right).

**Table 3. Recorded annual variation in pigs holdings and movements.** Observed variations (%) in the whole database and in the industry (in brackets).

|  | 2017–2018 | 2018–2019 | 2019–2020 |
|---|---|---|---|
| Number of active holdings | 1.12% (1.33%) | 0.82% (0.68%) | -3.64% (-3.52%) |
| Number of transports | -0.12% (0.00%) | -3.52% (-3.14%) | -13.52% (-12.74%) |
| Number of transported animals | -1.36% (-1.31%) | -0.21% (-0.07%) | -7.92% (-7.77%) |

**Table 4. Network statistics included in the forward stepwise process for exponential random graph model selection.** Network attributes in columns were associated with one to five exponential random graph terms (rows). Attributes and terms are described in Tables 5 and 6.

| | edges | distances | company | type | BRS | indus.sect | size | outdoor | insularity |
|---|---|---|---|---|---|---|---|---|---|
| edges | X | | | | | | | | |
| edgecov | | X | | | | | | | |
| nodefactor | | | X | X | X | X | X | X | X |
| nodeifactor | | | X | X | X | X | X | X | X |
| nodeofactor | | | X | X | X | X | X | X | X |
| nodematch | | | X | | | | | | |
| nodemix | | | | X | X | X | X | X | X |

growing networks, likely due to the three stages of production (birth/weaning; post-weaning; fattening) that allow for more trade along the production chain. As expected, the betweenness was higher in the whole network than in the within sector networks. The average densities were low, despite being higher in the breeding sector network. Except for the annual breeding network, the degree assortativity of all networks was negative (i.e. networks were

**Table 5. Description of network attributes.**

| Type | Attribute | Description |
|---|---|---|
| Edge attribute | Between-farm distances | Distance matrix per farm pair |
| Nodal attribute | Batch rearing system (BRS) | The batch rearing system (BRS) was also a qualitative variable with 8 levels based on the capacity of the farm and the most frequent interval of days between two imports (intD) over the 2016/2020 period as a proxy of the number of simultaneous batches in the rearing system: "4", "5", "7", "10", "small.10", "20", "small.20", "unidentifiable" |
| Nodal attribute | Company | Farms can be either independent or be part of a specific company |
| Nodal attribute | Insularity | The holdings located in Corsica were differentiated from those on the mainland |
| Nodal attribute | Outdoor housing system | Regarding the outdoor housing system, farms were divided into two categories: those with at least one outdoor rearing area and those with no outdoor rearing activity |
| Nodal attribute | Sector of the industry | According to their type, Farms can be part of the breeding sector (MU and NU) or of the growing sector (FA, FPW, FF, PW, PWF and FI). The indus. sect attribute was a 2-levels qualitative variable, basically synthetizing types attributes. According to the parsimony principle, this synthetic attribute aimed at reducing the number of variables in the final models |
| Nodal attribute | Size | Farm size is a qualitative variable with 3 levels based on the number of animals (nA): small, regular, large. For farms with a farrowing area, the number of sow was used as reference, however in the remaining, holdings with a post weaning area were categorised based on the number of growing pigs and finally finishers were categorised based on the number of barrows. For each type of holding, quantiles were calculated and size was attributed as follow: $nA < 25^{th}$ percentile: **small**; $25^{th}$ percentile $<= nA < 75^{th}$ percentile: **regular**; $nA >= 75^{th}$ percentile: **large** |
| Nodal attribute | Type of farm | The farm typology was a qualitative variable of 7 levels: specifying the type of production in the farm: farrow: "FA", post-wean: "PW", finish: "FI", farrow-to-finish: "FF", farrow-to-postwean: "FPW", postweaning-to-fininsh: "PWF", multiplier: "MU", nucleus: "NU" |
| Structural attribute | edges | Number of edges in the network |

**Table 6. Description of exponential random graph model terms used in the stepwise approach.**

| Type | ERGM terms | Description |
|---|---|---|
| Edge attribute-based | edgecov | Sum of quantitative edge attributes of all movements of the network |
| Nodal attribute-based | nodefactor | Number of times a farm with a given attribute is involved in a movement of the network |
| Nodal attribute-based | nodeifactor | Number of times a farm with a given attribute is the destination of a movement |
| Nodal attribute-based | nodeofactor | Number of times a farm with a given attribute is the origin of a movement |
| Nodal attribute-based (interaction) | nodematch | Uniform or differential homophily. The 'uniform homophily' is the number of movements in the network between two holdings having the same given attribute. All categories of the attribute are assumed to have the same propensity for within-group movements. The 'differential homophily' is calculated for each level of the attribute. It is the number of movements between two farms with the same given level of an attribute in the entire network (e.g the number of movements connecting two farms of a given company). Attribute levels are assumed to have independent propensities for within-group movements. |
| Nodal attribute-based (interaction) | nodemix | nodemix calculates, for each possible pairing of attribute levels, the number of edges in the network linking two farms with that pairing of values (e.g. number of movements from nucleus to farrow) |

disassortatives). The annual breeding network presented a positive degree assortativity indicating that farms were more often connected to nodes with similar number of trade partners. While being slightly higher in the breeding network than in the growing network, the reciprocity ratio remained low in all networks depicting rare bidirectional trade exchanges. The longer the studied time period, the higher the transitivity. While transitivity was higher in the breeding networks than in the growing ones, it also remained low in all networks ($< 4\%$). The average path length was higher in the whole networks, with farms being connected by three animal movements over a year on average, than in the breeding or the growing networks, where farms were connected by two animal movements over a year on average, which is consistent with the pyramidal structure of the industry. In all networks, almost 100% of the farms presented a Jaccard similarity coefficient equal to zero. Only the breeding networks presented a small amount ($<10\%$) of farms with a Jaccard similarity coefficient $> 0$. This coefficient was decreasing with the duration of the study period. The median distance of movements within the livestock sector was more than twice that of movements within the growing sector.

**Sows, piglets and growing pigs sub-networks characteristics** (Fig 4). The number of active farms involved in the transportation of growing pigs between farms was higher than those involved in piglets and sows movements, while the average number of movements was similar in all sub-networks. The number of active farms composing semestrial sub-networks was close to those of annual sub-networks. With identical numbers of movements, the growing pigs sub-network presented the highest number of isolated connected components while sows sub-network was composed of very few isolated connected components. Consequently, connected components in the sow's sub-network were slightly bigger than in other sub-networks. Median out-degrees were higher in networks built from transport of sows than in networks built from transport of piglets and growing pigs, and average betweenness was higher in networks built from transport of sows. The density was also higher in sows network than in piglets network, the latter presenting higher density than growing pigs network. The diameter of sow's sub-networks were slightly higher than in other sub-networks. Only the sub-networks based on sows transport presented a positive degree assortativity indicating that farms were

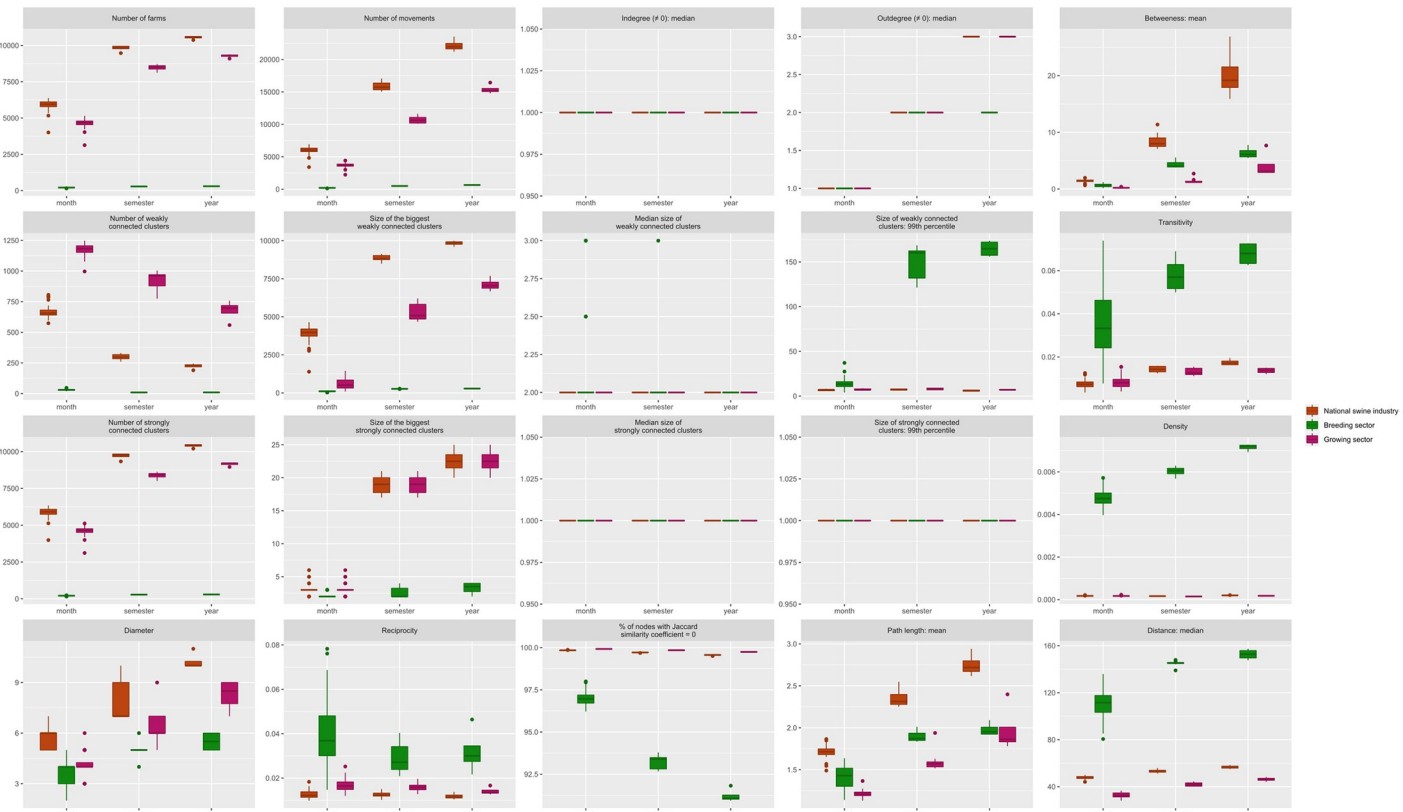

**Fig 3. Network centralities per production sector.** Average network centralities per month, semester and year for the whole industry, the breeding sector and the growing sector.

more often connected to farms with similar number of trade partners. Reciprocity was close to zero in all sub-networks, despite being relative higher in sow's and growing pig's sub-networks. As expected, transitivity was small in all sub-networks. The average path length was close to one in all networks, indicating that farms were connected by only one animal movements on average. Regardless of the considered period, most of the farms presented a Jaccard similarity coefficient equal to zero, especially in the growing pigs sub-networks. The median distance of sow movements was significantly longer than in piglets and growing pigs movements.

**Key drivers of movements**. The selected models for the sub-networks based on the animal transports contained 98, 70 and 96 significant co-variables respectively. 13 network statistics were common to all three sub-networks; they included edges, various effects of the different companies (mainly related to trade within companies), the specific movements from nucleus to farrow-to-finish farms and the involvement of post-weaners-to-finishers farms within the network (Table 7). Estimates for both the nucleus to farrow-to-finish movements and the implication of post-weaners-to-finishers farms in movements were negative in piglets and sows models and positive in growing pigs model, see S1–S3 Files. None of the selected models included the distance effect. None of the selected models included dyad dependent network statistics requiring MCMC estimation [32, 37]. The stepwise selection process started by drastically reducing the AIC and then reached a plateau before stopping at models with AICs of 50,869.05, 59,775.3 and 84,603.91 for the sows, piglets and growing pigs models respectively.

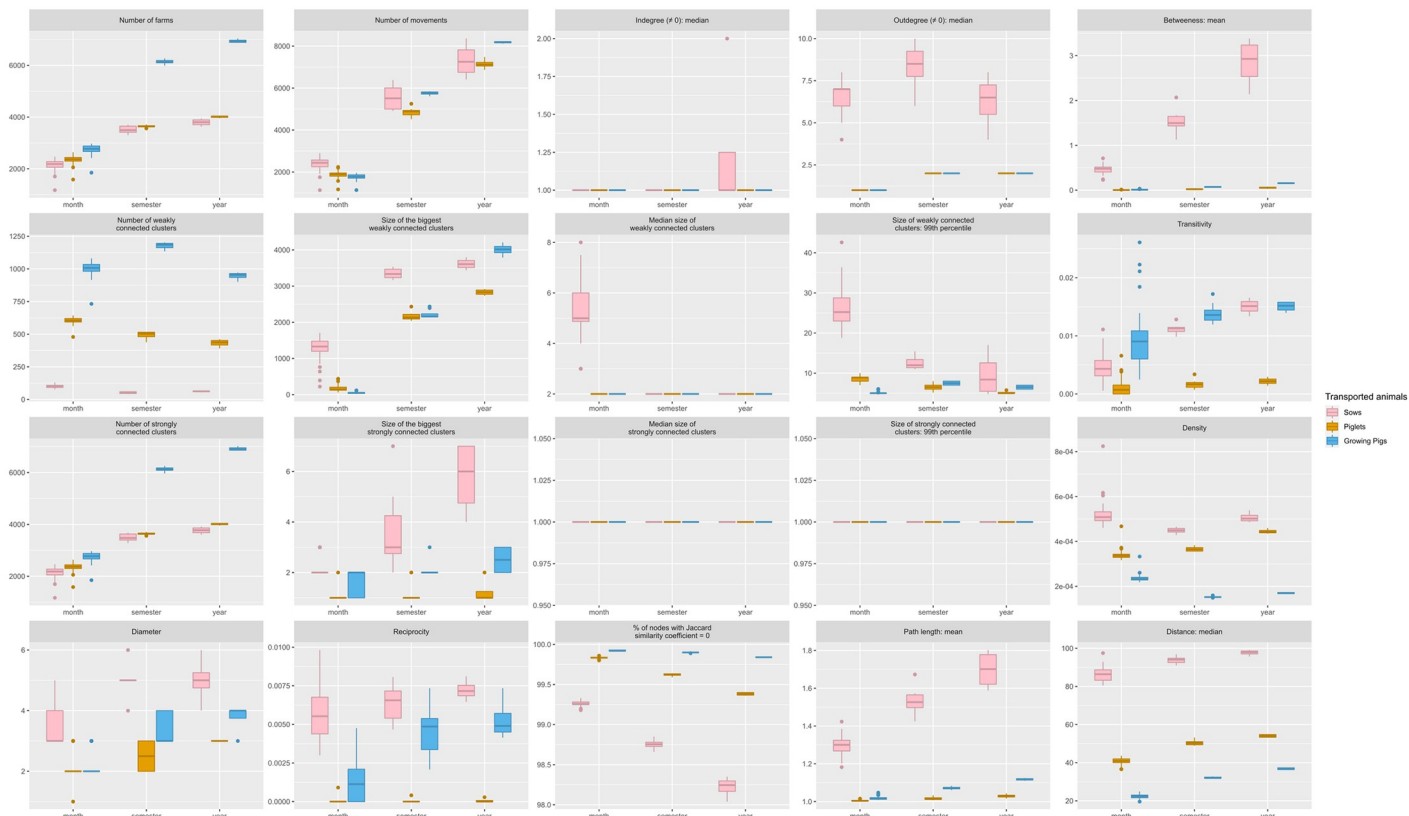

**Fig 4. Network centralities per type of transported animals.** Average network centralities per month, semester and year for the networks respectively built from the transport of sows, piglets and growing pigs.

For all three models, the selected variables correctly reproduced the global properties of the observed networks. The Goodness of fit (Gof) plots (Figs 5–7) showed that despite the absence of 'structural' statistics in the models (except for 'edges'), the observed network statistics (colored lines) were well captured by the distribution of the simulated network statistics values generated with the final ERGMs (underlying boxplots). While the models were fitted on the data of the second semester of 2019 (thick blue lines), the goodness of fit diagnostic showed a good fit with data of other semesters, whether they happened earlier (2017–2018) or later (2020) than the calibration data. Graphical comparison of observed and simulated networks maps also suggest a good fit of the models (Figs 8–10). According to the graphical Gof and maps study, the sow sub-networks had the best fit among three models. It was also the smallest sub-network.

**Sows**. Starting from the simplest model including only the edges and resulting in an AIC of 83,632.26, the stepwise forward procedure selected 98 network statistics among the 322 tested. 28,838 models were run and analyzed to obtain the best fit. The best fit was obtained using 98 covariables grouped into 18 different sets of term/attribute. The key variables involved edges structural attribute, breeding sector, company, outdoor housing system, types of farm, location on Corsica and type of batch rearing systems. The selection process allowed to reduce the AIC to 50,869.05. The cross-validation on the Gof showed a good adjustment of the model with distribution of simulated statistics mainly including the observed distribution (Fig 5). Visually, the mapping of the simulated networks was also very similar to the observed network (Fig 8).

**Table 7. Network statistics selected for each exponential random graph model.**

| Attribute | ERGM term | Sows model | Piglets model | Growing pigs model |
|---|---|---|---|---|
| edges | edges | X | X | X |
| Batch rearing system | nodefactor | X | X | X |
| | nodeifactor | | X | X |
| | nodeofactor | X | X | X |
| | nodematch | | X | X |
| | nodemix | X | X | X |
| Company | nodefactor | X | X | X |
| | nodeifactor | X | X | X |
| | nodeofactor | X | X | X |
| | nodematch | X | X | X |
| Insularity | nodeifactor | X | | |
| | nodemix | X | | X |
| Outdoor housing system | nodefactor | | X | X |
| | nodeofactor | | | X |
| | nodemix | X | X | X |
| Size of farm | nodeifactor | X | | |
| | nodeofactor | X | | X |
| | nodematch | | | X |
| | nodemix | X | X | X |
| Type of farm | nodefactor | X | X | X |
| | nodeifactor | | X | X |
| | nodeofactor | X | X | X |
| | nodemix | X | X | X |

The highly significant variables with the highest estimates were the breeding farms and finishers as senders, the trade between multipliers, the trade within the Companies 31 and 40 and the trade from nucleus to both multipliers and farrow-to-finish farms, see S1 File.

**Piglets**. Starting from the simplest model including only the edges and resulting in an AIC of 84,163, the stepwise forward procedure selected 70 variables among the 312 tested. 19,070 models were run and analyzed to obtain the best fit. The best fit was obtained using 70 covariables grouped into 18 different sets of term/attribute. The key variables also involved edges structural attribute, breeding sector, company, outdoor housing system, types of farm and type of batch rearing systems but not location on Corsica, all farms involved in piglets movements being located on the mainland, this was expected. The AIC of the selected model was equal to 59,775.3. Although the number of farms with high indegrees and outdegrees was slightly under-predicted, the cross-validation on the Gof still showed a good fit of the selected model (Fig 6). Visually, the mapping of the simulated networks appeared denser than the observed network, but the main patterns in the western side of the country were correctly reproduced by the ERGM (Fig 9). The highly significant variables with the highest estimates were the trade within companies (homophily) and especially within Company 37 and the trade from farrowers to both post-weaners and post-weaners/finishers, see S2 File. Edges variable was significant with a negative estimate, limiting the number of generated links.

**Growing pigs**. Starting from the simplest model including only the edges and resulting in an AIC of 111,648.9, the stepwise forward procedure selected 96 variables among the 334 tested. 27,252 models were run and analyzed to obtain the best fit. The best fit was obtained

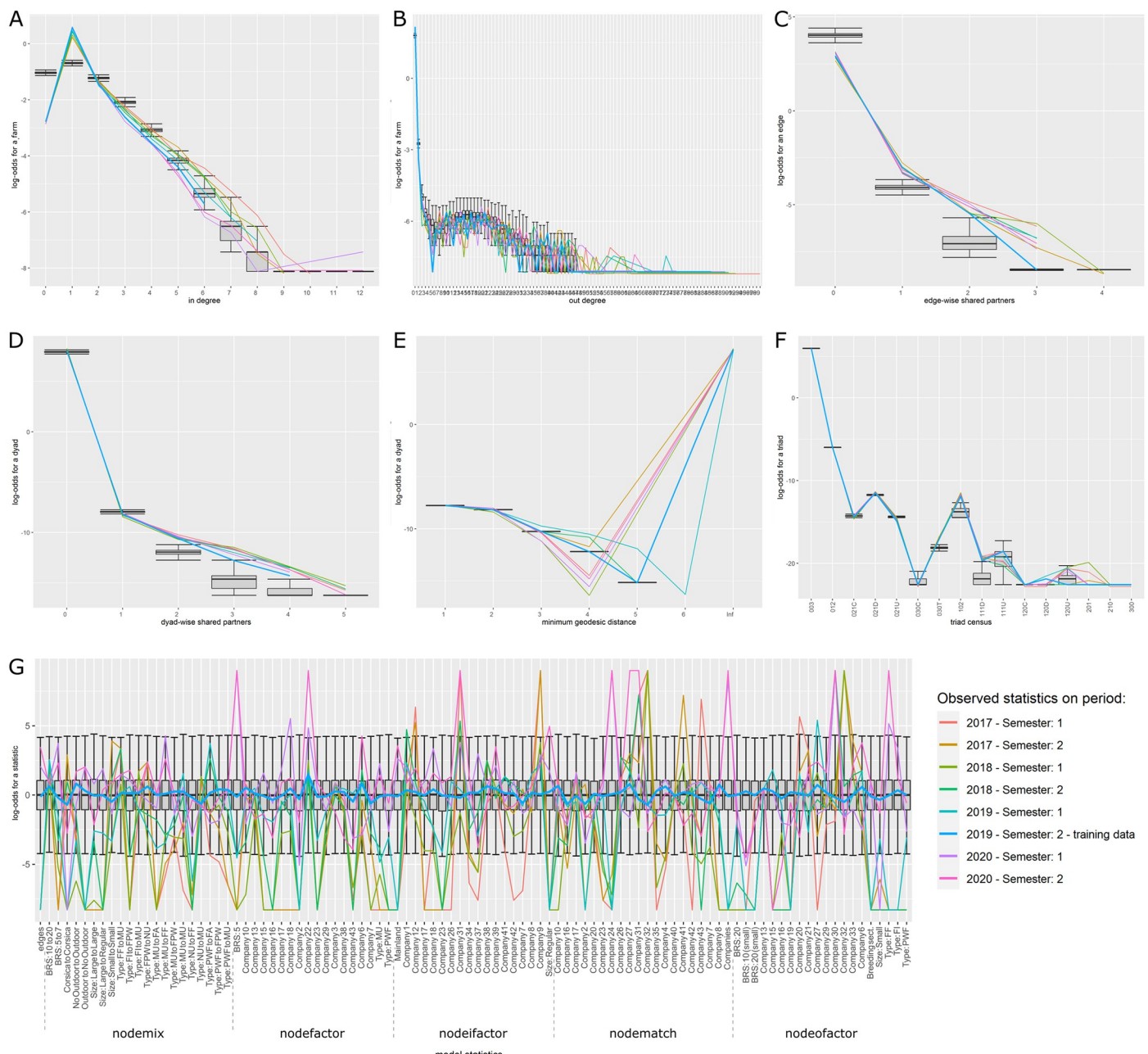

**Fig 5. Goodness of fit of the sows model.** Goodness of fit of the model adjusted for the transport of sows. Metrics of 4000 simulated networks (boxplot) are compared to metrics of observed semestrial network. Data of the second semester of 2019 were used for model fit (thick blue line). Colors correspond to different semesters' networks. Details of the network metrics: A: indegree distribution, B: outdegree distribution, C: edgewise shared partners distribution, D: dyadwise shared partners distribution, E: minimum geodesic distance distribution, F: triads census, G: performance of the final model on in-model statistics.

using 96 covariables grouped into 23 different sets of term/attribute. The key variables also involved edges structural attribute, breeding sector, company, outdoor housing system, types of farm, location on Corsica and type of batch rearing system. The AIC of the selected model was equal to 84,603.91. As for the piglet network, even though the distribution of simulated indegrees and outdegrees was slightly lower than the observed distribution, the cross-

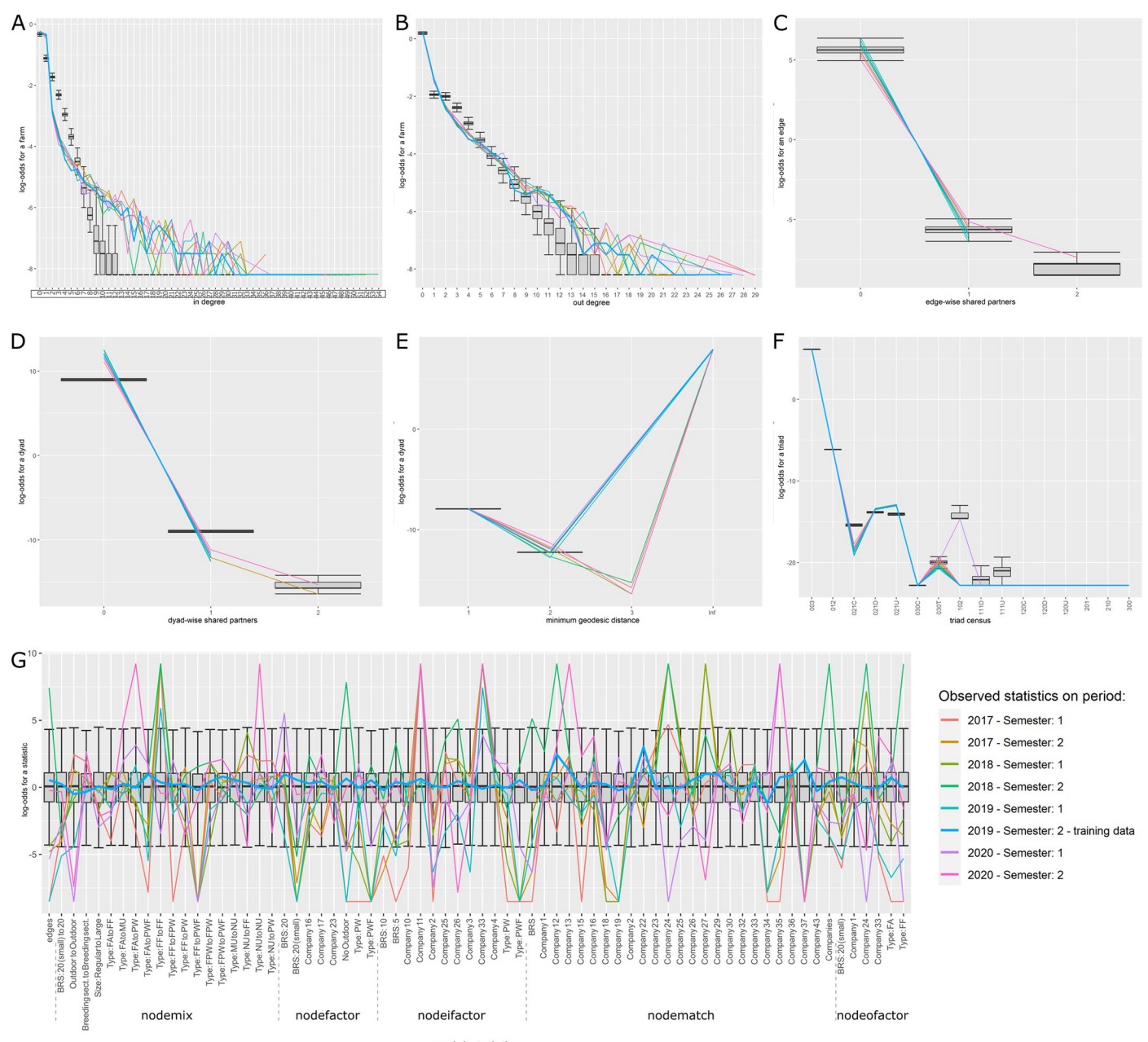

**Fig 6. Goodness of fit of the piglets model.** Goodness of fit of the model adjusted for the transport of piglets. Metrics of 5000 simulated networks (boxplot) are compared to metrics of observed semestrial network. Data of the second semester of 2019 were used for model fit (thick blue line). Colors correspond to different semesters' networks. Details of the network metrics: A: indegree distribution, B: outdegree distribution, C: edgewise shared partners distribution, D: dyadwise shared partners distribution, E: minimum geodesic distance distribution, F: triads census, G: performance of the final model on in-model statistics.

validation on the gof showed a good fit of the selected model (Fig 7). Visually, the mapping of the simulated networks seems to show that this is the least well fitted model with simulated networks slightly less structured than the observed network however once again the main patterns were correctly reproduced by the selected ERGM (Fig 10). The highly significant variables with the highest estimates were the trade within companies and especially within companies 5 and 13, the trade from farrow-to-finish farms to finishers, post-weaners/

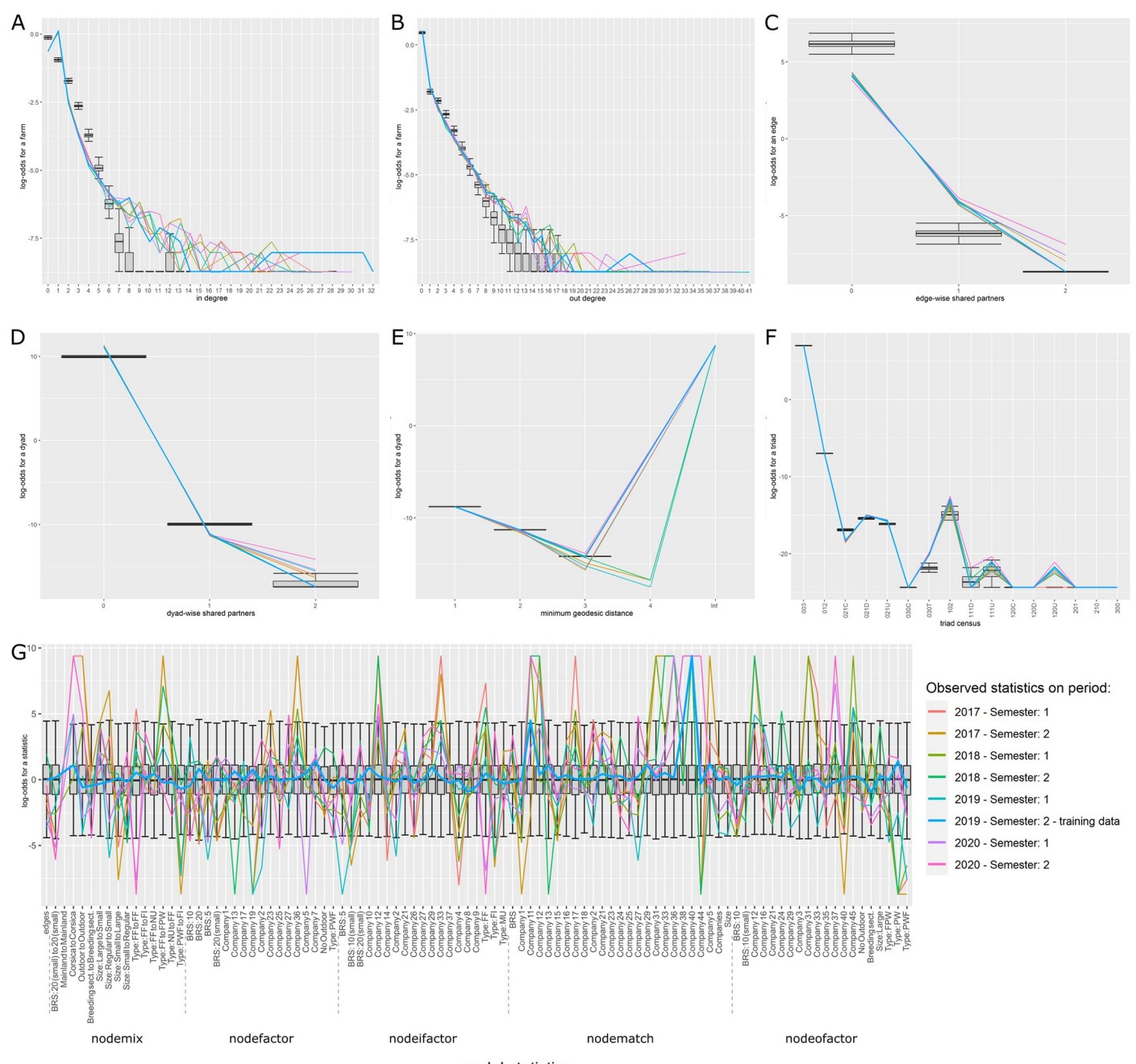

**Fig 7. Goodness of fit of the growing pigs model.** Goodness of fit of the model adjusted for the transport of growing pigs. Metrics of 6000 simulated networks (boxplot) are compared to metrics of observed semestrial network. Data of the second semester of 2019 were used for model fit (thick blue line). Colors correspond to different semesters' networks. Details of the network metrics: A: indegree distribution, B: outdegree distribution, C: edgewise shared partners distribution, D: dyadwise shared partners distribution, E: minimum geodesic distance distribution, F: triads census, G: performance of the final model on in-model statistics.

finishers, nucleus and farrow-to-finish farm and finally the status of farrower-post-weaners, post-weaners and more generally breeding farms as senders, see S3 File for more details. Like for piglet network, edges variable was significant with a negative estimate, limiting the number of generated links.

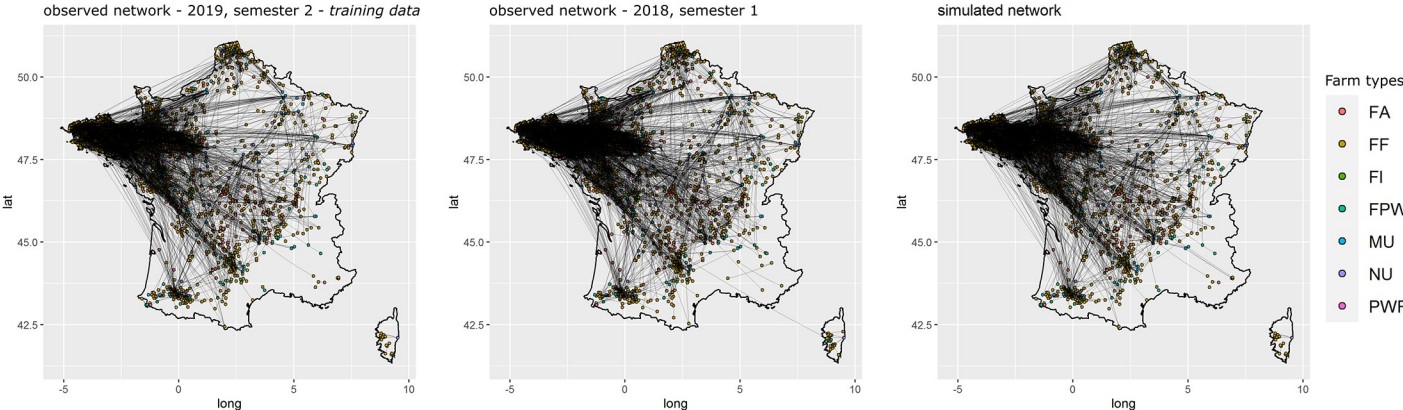

**Fig 8. Maps of observed and simulated networks for sows.** Mapping of semestrial networks drawn by the transport of sows. The network on the left was observed during the second semester of 2019, those data were used to fit the model, the network in the center was observed during the first semester of 2018, and the one on the right was simulated using the fitted exponential random graph model.

## Discussion

This study follows on from previous network analyses carried out for the 2014–2016 period [8]. Applying a similar approach, the present analysis was based on more recent data from the whole country covering a longer period of time. The various stages of data cleaning carried out over the years ensure quality of the data, as well as it corroborates the reliability of the results.

The overall outputs of the network analysis were in line with those of the period 2014–2016. The low levels of transitivity in the breeding and growing networks indicates a tree structure and the presence of hubs, along with the differential distribution of distances, it well reflects the pyramidal structure of the industry. However, there were some variations in specific patterns that deserved more consideration. The increasing number of active facilities observed between 2017 and 2019 was unexpected and could be related to recent efforts from database managers to inform and register all facilities/actors, as well as the cumulative amount of record over the years reducing the number of sites removed from the analysis because of missing

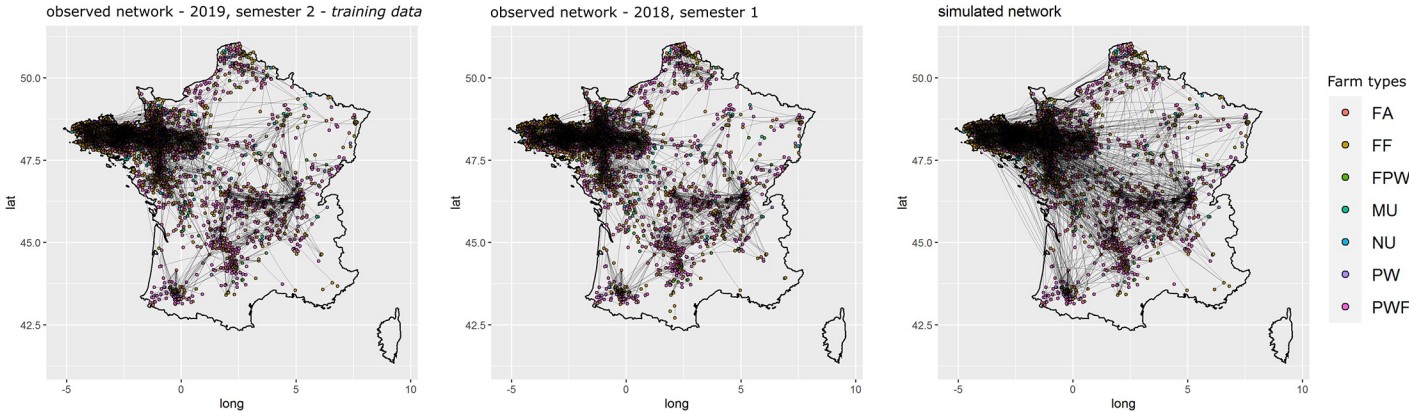

**Fig 9. Maps of observed and simulated networks for piglets.** Mapping of semestrial networks drawn by the transport of piglets. The network on the left was observed during the second semester of 2019, those data were used to fit the model, the network in the center was observed during the first semester of 2018, and the one on the right was simulated using the fitted exponential random graph model.

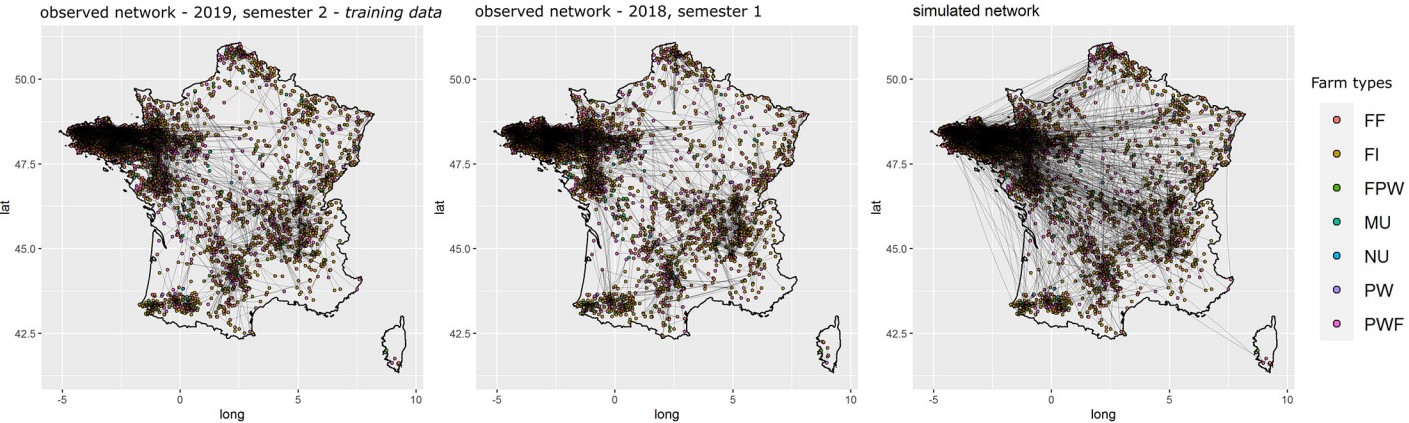

**Fig 10. Maps of observed and simulated networks for pigs.** Mapping of semestrial networks drawn by the transport of growing pigs. The network on the left was observed during the second semester of 2019, those data were used to fit the model, the network in the center was observed during the first semester of 2018, and the one on the right was simulated using the fitted exponential random graph model.

critical information. On the other hand, there was a sharp decrease in the number of active holdings in 2020. In 2020, the European pig production was strongly affected by both the Covid-19 crisis and the African Swine Fever pandemic [38], therefore some of the changes occurring in 2020 could be linked to those health crises [39–41]. The overall decrease in the number of movements and more generally in the number of transported animals confirms the general tendency observed over 2014–2016. Similarly, the average and median size of active farms per year remained stable from 2017 to 2019 and decreased slightly in 2020. The temporal-median degree illustrates some degree of loyalty between farms, at least on a yearly basis, as well as a prioritization of whole batch trading rather than disseminating animals among neighbouring farms for example. Median degrees of the network quantify the connectivity between farms and average betweenness measures the importance of a farm as an intermediary between different subnetworks. From an epidemiological perspective, the in-degrees and out-degrees respectively illustrate the capacity of a farm to be reached (infected) by other parts of the networks and to reach other parts of the network (and spread pathogens), and betweenness illustrates the vulerability of the nodes (the possibility of being reached at larger scale). As shown in Valdano *et al.* [42], stability of neighboring contacts increase the vulnerability of the node. Focusing on the disease spread potential of the network, both of them presenting small values, they support the hypothesis of an expected slow spread of pathogens via pig movements, with a slightly higher risk of spread when introduction occurs in multipliers or nucleus (higher median out-degrees in sows networks) [29, 43]. Moreover, the low levels of transitivity could reflect a diffuse circulation of pathogens in the network, without clustering effects. Temporal network and associated contact chains analysis could be performed to support this hypothesis. Applying classic social network analysis to animal movement data considering static aggregated networks already demonstrated its usefulness to identify the epidemiological hot spots and drivers. Pushing further to account for temporality in network topology could help disentangling spatio-temporal transmission patterns related to the production schedule.

Some spatial trade patterns seem to have changed since 2016. While the main partners countries remain the same European ones, the share of imports per country changed. While 47.3% of international imports came from Spain during the 2014–2016 period, it became much lower in 2017 representing only 5% of international imports and slightly increased over the studied period to reach 26% in 2020 to the benefit of Belgium, which increased its share of

French international imports from 33.3% over 2014–2016 to 57% over 2017–2020 [8]. Also, the distance distribution was different, with the average traveled distance halved for growing pigs and sows. The distances calculated from our datasets were smaller to those generally observed in other European countries such as Germany, Bulgaria or Spain than in 2014/2016 [25, 44, 45]. This may reflect a change in the network structure favoring short-distance exchanges. As recommended by other studies, using effective distances or travel-time instead of Euclidean distance could provide more accuracy on the actual distance traveled by trucks [46].

The annual number of active holdings (nodes) and movements (edges) were quite well represented by semestrial networks and no seasonal variation was detected. For all sub-networks, the centralities were similar, whether by month, semester or year, suggesting that the network structure was stable over time and that relatively short periods remain relevant to study network patterns. Those results are consistent with previous studies on static aggregation versus temporal approach to analyze European pig trade networks [3, 30]. Regarding the period under study, if the fitted ERGMs were used to define the probabilities of contact between farms in the context of epidemic surveillance (in "peace time"), the choice of the period would be sensitive in order to avoid over- or underestimating contacts. Used to feed an epidemiological model, over- or underestimating contact probabilities would lead to an over- or underestimation of the between-farm pathogen spread whether in space or time [47]. However, excluding the slaughterhouses from the analysis, we only observed inter-farms movements thus, according to the duration of the different growth stages, at least 180 days are required to observe complete pig production cycles and allow for most of the farms to be involved in inter-farms movements [30]. The period of one semester was chosen in order to allow the observation of several production cycles, (the growth period lasts 180 days) and the potential multiplicity of trading partners during the study period. This choice was all the more relevant as this period is commonly used in trade network analysis [3, 6, 48].

The high number of selected variables in all three ERGMs illustrates the inherent complexity of the trading patterns. Nevertheless among the multiplicity of selected variables, the most impacting ones are particularly relevant and expected. In the sows network, the influence of breeders (NU and MU) was clearly highlighted, as well as their privileged connection to FF. However, the position of FI as main senders in between-farm movements is less consensual and might be due to misinterpretation of some records in the database. In the same manner, the position of FA as main senders in the piglets networks was expected. To a lesser extent, other farrowing farms (FPW and FF) were also highlighted as significant senders. Finally, the key variables of the growing pigs networks, including breeders and especially FPW and PW as main senders and the privileged connection between FF and FI, illustrate the splitting of post-wean batches due to growing of pigs needing more space. It is also in line with a variable duration of fattening according to the animals implying potential sharing of batches between FI. In all three networks, the selection of homophily variables (*nodematch*) for companies reveals a certain form of clustering within companies, this was expected due to specific partnership and contract through the company. The companies highlighted in the three networks were different, indicating a specialization in the production stages. Companies 31 and 40 seemed specialized in breeding activities both absent from variables selected for the piglets networks but involved as main senders in the growing pigs networks. Company 37 seems specialised in farrowing/post-weaning without being selected in the sows network variables and being highlighted as a main exporter in the growing pigs networks. Finally, companies 5 and 13 seems specialised in post-weaning/finishing both selected as non-significant variables in the sows networks and the company 13 only selected to favour the within company exchanges in the piglets network. Those companies might be targeted by surveillance and control activities

to reduce the risk of introducing contaminated animals into the food chain. In the growing pigs and piglets networks, as expected farms organized as a 10-BRS or a 20-BRS were more involved in the movements. The movements of piglets and growing pigs seemed to be privileged between farms with the same BRS, that could be explain by a synchronisation of the production stages. In the sows networks model, expect for the connection from 10-BRS to 20-BRS that was weakly significant, all other variables related to BRS had negative estimates limiting the implication of 10-BRS and 20-BRS as senders. Also, in all models, the variables associated to the 5-BRS had negative estimates. Since BRS is estimated from movement frequency and farm size, these results should be viewed with caution, and should be confirmed using data with a validated BRS. In both piglets and growing pigs networks, the traditional production systems (without outdoor areas) were more involved in the movements especially as senders in the growing pigs networks. In the piglets and sows networks, interactions were favoured between farms with outdoor housing (positive estimates in piglets networks for this interaction and negative estimates for the exchanges between farms with traditional housing and outdoor housing in the sows network). This was expected. In this study, a 2-categories factor was used nevertheless, the relations could be more detailed including specific production systems such as straw farming or organic farming. Size effect were specific to each model, with an expected homophily effect in the sows and growing pigs networks, as well as some specific interactions between sizes favoured or hindered in all three models. We also observed an expected prioritization of movements from larger to smaller farms in piglets and growing pigs networks. This pattern is not observed in sow networks, which is not surprising since sows in batches are generally culled and renewed over time and not by full batches. The insularity effect was only significant in the growing pigs networks, favouring the within island exchanges. Given the small number of farms on the island, the companies and type of farm effects may be sufficient to predict island movements.

Euclidean distances or other spatial attributes such as the region, are generally identified as a key factor responsible for movements [24, 25, 29, 44], however the present results reveal that, it might not be mandatory to correctly represent the spatial structure of the pig networks. In France, the supply chain between agricultural companies being already spatially organized, it seems sufficient to represent its spatial structure. This highlights the need for a prior descriptive analysis of the network to identify objective characteristics that could explain the structure of contacts. All others *a-priori* selected attributes were involved in at least two sub-networks. This suggests that our prior selection was accurate, even if more features could probably be added to the models, such as road density used in other studies [25]. While the stepwise selection does not consider all possible combinations of potential predictors, the two-ways approach improves variable selection and provides a robust fit. Performing the stepwise selection over another semester could also support the predictor selection. However, the Gof, cross-validation and graphical validation showed that these variables were already able to correctly fit the models to the observed networks. The identification of factors responsible for the choice of trading partners revealed and confirmed the important role of companies, type of farming, farm size, outdoor housing systems and BRS in the French between-farm network. The selection of BRS attribute as a key driver of movements corroborates the classification that was made. By highlighting the specific relationship related to BRS in the three sub-networks, this study reveals the importance of this factor and the relevance of including it in further data collection. Integrating structural statistics related to local structures and clustering, such as geometrically weighted edgewise shared partner (GWESP), geometrically weighted dyadic shared partner (GWDSP), or geometrically weighted degree (GWD), could also improve the prediction, however the low levels of transitivity seems to reflect low clustering effect and the variables related to dyad-dependent centralities seems not yet adapted to networks as big as

the one analyzed in this study [36]. ERGMs are still in development and, although they are increasingly used in social network studies, their use in animal health remains uncommon [5, 24, 25, 49–52]. However, they provide better knowledge of network structure [23, 32, 53, 54] and, although fitting ERGMs explaining large networks is particularly challenging, time-consuming and requires dedicated powerful computer clusters [32, 34, 35, 53, 55], it eventually provides relevant information on key factors of farm-to-farm contacts. Allowing to understand complex networks, ERGMs seem to be a relevant tool to simulate more realistic probabilities of contact between farms and consequently, the probability of disease spread in various epidemiological contexts.

The division of the general network into three sub-networks based on growth stages is probably responsible for the relatively low values of the minimum geodesic distance, the triad census distribution, the edge-wise shared partners and the dyad-wise shared partners in each sub-network. For a more accurate validation, a demographic model describing animal movements in the meta-population formed by the industry's farms should be developed. The contact probabilities estimated from the fitted ERGM could feed such demographic model greatly improving its predictive capabilities without further need of observed data. The resulting simulated network would probably present more heterogeneity in contact patterns, especially in the type of triad and geodesic distances distribution. Analyzing those simulated networks using dynamic network approaches would highlight the central farms (e.g. those with persistent highest betweenness, outdegree or indegree) and provide relevant information to guide national veterinary services designing network-based surveillance, prevention, and control interventions [56, 57]. In addition, it could be used to compare simulated and observed contact chains with tools such as EpiContactTrace [58] and support the validation of current models. To go even further, the outputs of such model could be used to feed multi-levels epidemiological models, such as the one developed by [59] using SimInf [60], to compare preventive and control strategies [61]. In case of foodborne diseases contracted through the consumption of contaminated pork, such as hepatitis E, a better understanding of the movement of live animals in the production chain could help to estimate the circulation of pathogens in the pig industry and, consequently, to better assess the risk of introduction in the human food chain. Such tools can significantly contribute to forecasting epidemics and improving epidemiological surveillance and control.

## Conclusion

This study is one of the first to use ERGMs to understand and predict the pig exchanges within the production chain at the scale of a whole country (except for the overseas territories). Better understanding farmer's choices for trading partners in peace time at the national scale could better inform French policy makers on transmission routes of pathogens that silently circulate in swine industry.

CNA methods provided a better knowledge of the network structure [8, 12, 18, 62] and, even though the fit of ERGMs explaining large network is particularly challenging [34, 55], ERGMs ends up to provide relevant information on key drivers of between-farm contacts.

ERGMs were used to analyze and predict the structure of networks built by the between-farm movements of live animals in the French pig industry. This study focuses on the sub-networks formed by the movements of sows, piglets and growing pigs. The analysis of these networks improves our understanding of the factors affecting farmers' choices in terms of farming partnerships. Outputs provided in this study could certainly contribute to feed predictive epidemiological model of between-farm disease transmission such as [59]. Rightly designed, simulated networks can be used to feed spatio-temporal simulation tools such as

epidemiological models, significantly improving their predictive capacities, they could provide essential information on transmission routes and better inform French policy makers on prevention and control of swine diseases in France [56, 63, 64].

## Supporting information

**S1 Table. Montly time-series of centralities for the national swine industry network.**
(TIF)

**S2 Table. Semestrial time-series of centralities for the national swine industry network.**
(TIF)

**S3 Table. Annual time-series of centralities for the national swine industry network.**
(TIF)

**S4 Table. Montly time-series of centralities for the within-breeding sector network.**
(TIF)

**S5 Table. Semestrial time-series of centralities for the within-breeding sector network.**
(TIF)

**S6 Table. Annual time-series of centralities for the within-breeding sector network.**
(TIF)

**S7 Table. Montly time-series of centralities for the within-growing sector network.**
(TIF)

**S8 Table. Semestrial time-series of centralities for the within-growing sector network.**
(TIF)

**S9 Table. Annual time-series of centralities for the within-growing sector network.**
(TIF)

**S10 Table. Montly time-series of centralities for the sows movements.**
(TIF)

**S11 Table. Semestrial time-series of centralities for the sows movements.**
(TIF)

**S12 Table. Annual time-series of centralities for the sows movements.**
(TIF)

**S13 Table. Montly time-series of centralities for the piglets movements.**
(TIF)

**S14 Table. Semestrial time-series of centralities for the piglets movements.**
(TIF)

**S15 Table. Annual time-series of centralities for the piglets movements.**
(TIF)

**S16 Table. Montly time-series of centralities for the growing pigs movements.**
(TIF)

**S17 Table. Semestrial time-series of centralities for the growing pigs movements.**
(TIF)

**S18 Table. Annual time-series of centralities for the growing pigs movements.**
(TIF)

**S1 File. ERGM summary for sows network.**
(PDF)

**S2 File. ERGM summary for piglets network.**
(PDF)

**S3 File. ERGM summary for growing pigs network.**
(PDF)

## Acknowledgments

We would like to thank the comprehensive French national pig identification database
(BDporc) for providing the data.

## Author Contributions

**Conceptualization:** Mathieu Andraud.

**Formal analysis:** Pachka Hammami.

**Funding acquisition:** Stefan Widgren, Nicolas Rose, Mathieu Andraud.

**Investigation:** Mathieu Andraud.

**Methodology:** Pachka Hammami, Stefan Widgren, Vladimir Grosbois, Andrea Apolloni, Nicolas Rose, Mathieu Andraud.

**Supervision:** Nicolas Rose, Mathieu Andraud.

**Validation:** Pachka Hammami, Vladimir Grosbois, Andrea Apolloni, Nicolas Rose, Mathieu Andraud.

**Visualization:** Pachka Hammami.

**Writing – original draft:** Pachka Hammami.

**Writing – review & editing:** Pachka Hammami, Stefan Widgren, Andrea Apolloni, Nicolas Rose, Mathieu Andraud.

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
