## [Decision Letter · Decision Letter 0]

4 Nov 2021

PONE-D-21-29344Complex network analysis to understand trading partnership in french swine productionPLOS ONE

Dear Dr. Hammami,

Thank you for submitting your manuscript to PLOS ONE. After careful consideration, we feel that it has merit but does not fully meet PLOS ONE’s publication criteria as it currently stands. Therefore, we invite you to submit a revised version of the manuscript that addresses the points raised during the review process.

Both reviewers have raised some important concerns regarding clarity in the presentation of the results. Please, consider revising the paper to carefully address those concerns. 

We look forward to receiving your revised manuscript.

Kind regards,

Michele Tizzoni

Academic Editor

PLOS ONE

“This work was supported by funding from the European Union’s Horizon 2020 Research and Innovation programme under grant agreement No 773830: One Health European Joint Programme.”

We note that you have provided additional information within the Acknowledgements Section. Please note that funding information should not appear in the Acknowledgments section or other areas of your manuscript. We will only publish funding information present in the Funding Statement section of the online submission form.

 “This work was supported by funding from the European Union’s Horizon 2020 Research and Innovation programme under grant agreement No 773830: One Health European Joint Programme.  The funders had no role in study design, data collection and analysis, decision to publish, or preparation of the manuscript.”

Reviewers' comments:

Reviewer's Responses to Questions

**Comments to the Author**

1. Is the manuscript technically sound, and do the data support the conclusions?

Reviewer #1: Yes

Reviewer #2: Yes

2. Has the statistical analysis been performed appropriately and rigorously? 

Reviewer #1: Yes

Reviewer #2: Yes

3. Have the authors made all data underlying the findings in their manuscript fully available?

Reviewer #1: No

Reviewer #2: Yes

4. Is the manuscript presented in an intelligible fashion and written in standard English?

Reviewer #1: Yes

Reviewer #2: Yes

5. Review Comments to the Author

Reviewer #1: A very interesting paper utilising a comprehensive dataset. The paper is well written and does a good job at describing the complex methods used. The discussion provides some insightful suggestions for future work.

I’m interested in why the authors chose to look at breeding and growing separately, considering the pyramidal nature of the industry, can they please justify this. Why wasn’t an ERGM done on the whole industry?

I would have liked to seen more investigation about the effects of companies. Do you think this might influence disease transmission in the French pig industry? Have you looked at company specific networks or run any community algorithms on this? Company networks might reveal higher clustering coefficients, c.f. lines 294-295, which may have an impact on disease spread. Please discuss this, or if you feel it has no impact, please justify why it is not relevant in this study.

Specific queries

You mention in the results (lines 313-318) that degree and betweenness were higher in the breeding sows networks. Why was this? Does the breeding sows definition include gilts/nulliparous female pigs? If so, please clarify this or suggest what the type of movements are in this network.

Boar stations are an epidemiological dead end – please can you define a boar station to clarify this.

Please can the authors briefly recap the key findings of ref 8.

In figures 3 and 4, can the authors report degree divided into in-degree and out-degree as this is particularly relevant in characterising the breeding vs growing networks.

In the results you mention the growing network had variable biggest cluster (giant component?) sizes among semesters (lines 283-284), yet in the discussion (lines 442-446) you surmise that there are no seasonal fluctuations in centrality and don’t mention clustering. Please can you expand on this.

Line 284 – you mention high average degree for breeding premises, but presumably this is out-degree only? Can you please clarify this as it is an important distinction for disease transmission.

Have the authors included the coefficients for the ERGMs? I would like to see a table for this. I think they are important for understanding the networks and would like to see the relative impact of these variables as referenced in the discussion (lines 473-476).

L414-416 – please provide some more evidence for this statement and back it up with references that have shown this.

Suggestions (not required for acceptance)

Figure 1 –it is difficult to locate the abbreviations for these nodes, please make them clear. For the connection between FF and FI you can’t see the edge – could it be rearranged so we can see how many movements? Does the thickness of the arrows correspond to the number of animals in plot on the left as well? They all seem the same thickness. Please explain more in the figure captions.

Fig 3 – some of the colours are difficult to see with the boxplots outlines in black. It might be helpful to change them to the same as the fill colour.

Table 7 – It would be helpful to add the details in brackets from table 5 into the attribute column to make it easier to read.

Although there are restrictions on the underlying data, it would be useful for the code used for the analysis to be made publicly available.

Reviewer #2: In their paper "Complex network analysis to understand trading partnership in french swine production" the authors analyze the french swine trade network. Besides the classical network analysis approach, they use an exponential random graph model (ERGM) in order to find the key drivers forming the network. The main result is that the networks' properties can somehow be captured be the ERGM.

I like the idea that an ERGM is applied to such a system to find key drivers, and in particular including the different production types explicitly. On the other hand, these reader is left with no clear idea what these key drivers are actually. An overview of this result would strongly increase the quality of the papers main statement.

Although the paper contains a relatively large number of linguistic errors, it is in general well written. The results section would benefit from proof reading.

I recommend publication of the paper after a major revision.

# General comments

The phrase "premise" (singular) appears multiple times in the manuscript. The plural premises is correct in this context, however, the singular "premise" means something else and must be replaced by "farm", "premises", or similar. Please check the whole document (including figure axes) accordingly.

Tables 4, 5, 6 are very hard to follow. The information contains massive nested explanations and this must be improved in order to be readable.

# Specific comments

Table 1, footnote

"This approximation was validated...". Justified instead of validated.

Table 1, caption

Premise. See my comment above.

l 73

premise. See my comment above.

l 75

',' after the ')'

Table 2

The word 'cluster' is not precise. It can mean many things and you certainly mean components. If so, be clear about it, i.e. strongly/weakly connected components etc.

Table 2

Typo "number of groups of premises disconnected from each other".

Table 2

"Assortativity degree" probably means "degree assortativity" or "Assortativity by degree"?

Table 2

Transitivity. This is defined for undirected networks only. In the directed case, there are more possibilities for triangular motifs. Explain.

Table 2

What does "prop.nodes.Jaccar SC = 0" mean?

Table 2

Average distance. Be clear that the distance here is geographical, as opposed to the network distances.

l 158

08 kg -> 8 kg

l 165

ERGMs are not modified logistic regressions! They define an ensemble and new networks can be sampled from this ensemble. This is something different from a regression. Clarify.

l 172

Typo: Model selection.

l 175

Use the correct letters for the names Erdos and Renyi. (The former are not available for this plain text review).

l 188

MCMLE. Not explained. Also other (even though common) abbreviations, such as MCMC or MLE, should be written out when used for the first time.

l 256

Typo: Analysis

l 288

Assortativity can be computed for different properties. I guess you mean the degree assortativity here?

l 293

The longer..., the higher...

Table 5

Isn't the "sector of the industry" attribute redundant, when the "type of premises" attribute is known? Please provide a brief explanation.

Table 5

Typos: 25th and 75th percentile.

l 317

"himself" -> "the latter"?

Table 6

Row 1: Does '... for each movements appearing...' mean 'for each movement pair'?

Table 6

Typo: e.g. (last row).

l 323

Typo: ... all sub-networks.

l 324

'All network' should be replaced by the whole network or so.

l 324

typo: on average.

l 337

growing pics models.

l 334

Write out 'goodness of fit (Gof)' at first appearance.

Figures 5 and following

Figures must be subdivided, i.e. Fig. 5 A, B, C, etc. in order to habe clear references on the specific results.

(And the figure axis contains 'premise', see my general comment).

Figures 5 and following

The ordering of the figures is confusing. Consider either Gof followed by map or place 8, 9, 10 later.

l 386

so -> though?

ll 417 ff

"nevertheless temporal network..." This is not related to the statement on clustering before. It should be related to the next sentence instead. Indeed, temporal and weighted networks are very interesting in this context and would give better results. However these sentences have no logical order and seem arbitrary. Please rephrase.

l 442

was detected.

l 484

typo dependent.

l 529

The term 'stochastic network' means something else. Please rephrase.

6. PLOS authors have the option to publish the peer review history of their article (what does this mean?). If published, this will include your full peer review and any attached files.

Reviewer #1: No

Reviewer #2: No

---

## [Author Response · Author response to Decision Letter 0]

14 Feb 2022

Response to the reviewers

We wish to thank the editor and reviewers for their critical assessment of our work, their comments, suggestions, and corrections, which have helped us to greatly improved the manuscript. In the following, we address reviewers concerns point by point, quoted lines corresponding to the new pdf version without track changes.

We apologize for the Table 5 format in the track change version, modifications are not well handled by LaTeX producing a huge and unreadable table. We recommend to look at the revised pdf version.

Reviewer 1

Reviewer Point P 1.1 — I’m interested in why the authors chose to look at breeding and growing separately, considering the pyramidal nature of the industry, can they please justify this. Why wasn’t an ERGM done on the whole industry?

Reply: In general, between-farm contacts could be described by fitting an ERGM to the whole industry, nevertheless sows, piglets and pigs movements involve different types of farms and different duration of stay in production sectors; therefore, we assumed they were submitted to different drivers. As, the type of animals cannot be included in edge characteristics, because a unique value is required for all potential movements/pairs of farms, working on specific sub-networks seemed a more relevant approach.

Reviewer Point P 1.2 — I would have liked to seen more investigation about the effects of companies. Do you think this might influence disease transmission in the French pig industry? Have you looked at company specific networks or run any community algorithms on this? Company networks might reveal higher clustering coefficients, c.f. lines 294-295, which may have an impact on disease spread. Please discuss this, or if you feel it has no impact, please justify why it is not relevant in this study.

Reply: The study aimed to understand interactions within and between companies at national scale. We fully agree with you on the important role of companies in the french industry. In preliminary analyses, we observed that communities identified using random walks on the network adequately fitted the main french companies. On the other hand, small companies and independent farms were not represented. Companies were identified as one of the driving variables for all three subnetworks, showing their importance on the structure. Including companies as explanatory variables is essential to explore the contact pattern between company-based communities, which are in turn important to analyse the spread of an infectious agent at national scale. Specific companies influences on between farms movements were described and discussed in the results (L.365-412) and the discussion (L.503-513) parts.

Specific queries

Reviewer Point P 1.3 — You mention in the results (lines 313-318) that degree and betweenness were higher in the breeding sows networks. Why was this? Does the breeding sows definition include

gilts/nulliparous female pigs? If so, please clarify this or suggest what the type of movements are in this network.

Reply: Indeed, sows include gilts and nulliparous female pigs. Therefore, higher degrees (mainly outdegree) and betweeness were expected in the breeding sows networks as nucleus farms supply female pigs to multipliers, who feed multiple production farms. The sows group definition was completed (L.157) and the word ”sows”, more generic, was used instead of ”breeding sows”.

Reviewer Point P 1.4 — Boar stations are an epidemiological dead end – please can you define a boar station to clarify this.

Reply: L119: The synonym: “insemination centre” was given in parentheses when mentioned for the first time (L.118). Details were also added (L.160-166).

Reviewer Point P 1.5 — Please can the authors briefly recap the key findings of ref 8.

Reply: In reference 8, two network structures were analysed including or not the transit of trucks in farms in the absence of animal exchange. The analysis of these two networks allowed to identify specific features which need to be accounted for when using animal movements in epidemiological assessment. Indeed, different pathogens will deserve different network representation. Hepatitis E virus being transmitted though direct contacts, and ingestion of contaminated feces, requires physical contacts or shared environments, which correspond to the ’Animal introduction model’. But one may think of an airborne pathogen (e.g. influenza viruses) which may spread from a truck, containing infected individuals, and present in a farm to load a batch of animals. In that case a transmission could be possible from the truck to the herd, and a ”Transit model” needs to be implemented.

We do not feel adding such details in our manuscript, focusing on animal introduction model, would be of any help to the reader, the reason why we decided not to.

Reviewer Point P 1.6 — In figures 3 and 4, can the authors report degree divided into in-degree and out-degree as this is particularly relevant in characterising the breeding vs growing networks.

Reply: Figure 3 and 4 were updated discriminating indegrees and outdegrees. Table 2 descriptions were updated as well.

Reviewer Point P 1.7 — In the results you mention the growing network had variable biggest cluster (giant component?) sizes among semesters (lines 283-284), yet in the discussion (lines 442-

446) you surmise that there are no seasonal fluctuations in centrality and don’t mention clustering. Please can you expand on this.

Reply: Figure 3 and 4 were updated discriminating strongly and weakly connected components. De- scription in Table 2 were updated as well. Although the size of the largest weakly connected component (WCC) varies slightly more between semesters than from year to year, the variability in strongly con- nected component (SCC) size appears stable between semesters and years. We attributed this slight difference to a sample size effect (only 8 semesters and 4 years) and removed the mention from the re- sults. Furthermore, the observed differences could not be attributed to a seasonal effect. In comparison to other production system, (eg. foie-gras ducks), no seasonality has been observed in swine commercial movements, as documented in several papers (eg. Refs 4, 8, 24, 28, 30).

Reviewer Point P 1.8 — Line 284 – you mention high average degree for breeding farms, but presumably this is out-degree only? Can you please clarify this as it is an important distinction for disease transmission.

Reply: Following on P 1.6, as average degrees were discriminated between in-degrees and out-degrees, keeping only temporal-median values, the results and discussion were adapted, particularly by notifying the higher spread potential of nucleus and multiplier farms based on the higher out-degrees observed in sows networks (L.439-448).

Reviewer Point P 1.9 — Have the authors included the coefficients for the ERGMs? I would like to see a table for this. I think they are important for understanding the networks and would like to see the relative impact of these variables as referenced in the discussion (lines 473-476).

Reply: We want to thank both reviewers for pointing this out. As mentioned in response to P 2.1, we added a description of the selected models including parameters and associated coefficients in supporting information (S1-3 Files). Key parameters were detailed (L.345-349 ; 365-412) and discussed (L.487-538) for each model.

Reviewer Point P 1.10 — L414-416 – please provide some more evidence for this statement and back it up with references that have shown this.

Reply: The statement was developed and two reference on studying disease spread based on network patterns were added (ref 43-44).

Suggestions (not required for acceptance)

Reviewer Point P 1.11 — Figure 1 –it is difficult to locate the abbreviations for these nodes, please make them clear. For the connection between FF and FI you can’t see the edge – could it be rearranged so we can see how many movements? Does the thickness of the arrows correspond to the number of animals in plot on the left as well? They all seem the same thickness. Please explain more in the figure captions.

Reply: Abbreviation were added to the legend. After reviewing the figure, we spotted an error in the right Figure, it has been updated.

Reviewer Point P 1.12 — Fig 3 – some of the colours are difficult to see with the boxplots outlines in black. It might be helpful to change them to the same as the fill colour.

Reply: We would like to keep the median visible, nevertheless, while keeping the black borders, the new Figure is more readable.

Reviewer Point P 1.13 — Table 7 – It would be helpful to add the details in brackets from table 5 into the attribute column to make it easier to read.

Reply: Attribute names were simplified and homogenized between tables 4 - 7.

Reviewer Point P 1.14 — Although there are restrictions on the underlying data, it would be useful for the code used for the analysis to be made publicly available.

Reply: At this stage, the code is very french production specific. A package including example data, demographic and epidemiological models will be build and made available to public in the near future on GitHub.

Reviewer 2

Reviewer Point P 2.1 — I like the idea that an ERGM is applied to such a system to find key drivers, and in particular including the different production types explicitly. On the other hand, these reader is left with no clear idea what these key drivers are actually. An overview of this result would strongly increase the quality of the papers main statement.

Reply:

We want to thank both reviewers for pointing this out. As mentioned in response to P 1.9, we added a description of the selected models including parameters and associated coefficients in supporting information (S1-3 Files). Key parameters were detailed (L.345-349 ; 365-412) and discussed (L.487-

538) for each model.

Reviewer Point P 2.2 — Although the paper contains a relatively large number of linguistic errors, it is in general well written. The results section would benefit from proof reading.

Reply: We apologize for this and have taken steps to improve the English of the document.

General comments

Reviewer Point P 2.3 — The phrase ”premise” (singular) appears multiple times in the manuscript. The plural premises is correct in this context, however, the singular ”premise” means something else and must be replaced by ”farm”, ”premises”, or similar. Please check the whole document (including figure axes) accordingly.

Reply: ——- Note for myself: Check the figures before resubmitting ———– Thanks for noticing, it has been modified. The words ’premises’ and ’premise’ were replaced by ’holding(s)’ when including farms, slaughterhouses, rendering plant, etc., and by ’farm(s)’ when only referring to the production units.

Reviewer Point P 2.4 — Tables 4, 5, 6 are very hard to follow. The information contains massive nested explanations and this must be improved in order to be readable.

Reply: We totally agree that such tables are difficult to read. However, it appears essential to clearly define the terms employed in the ERGM modelling framework. As for network terms, the vocabulary is highly specific and deserve, in our opinion, to be explicitly defined. Networks attributes in Table 5 were simplified and reorganized by alphabetical order (Type and Attribute), as well as attribute in Table 7. The best way we found was to design tables where each term can be relatively easily caught.

Specific comments

Reviewer Point P 2.5 — Table 1, footnote ”This approximation was validated...”. Justified instead of validated.

Reply: The correction has been made.

Reviewer Point P 2.6 — Table 1, caption Premise. See my comment above. The correction has been made.

Reply: The correction has been made.

Reviewer Point P 2.7 — l 73 premise. See my comment above.

Reply: The correction has been made.

Reviewer Point P 2.8 — l 75 ’,’ after the ’)’

Reply: The correction has been made.

Reviewer Point P 2.9 — Table 2 The word ’cluster’ is not precise. It can mean many things and you certainly mean components. If so, be clear about it, i.e. strongly/weakly connected components etc. Weakly connected components of a graph

Reply: The word ’cluster’ was replaced by connected components. Associated statistics were detailed discriminating strongly/weakly connected components.

Reviewer Point P 2.10 — Table 2 Typo ”number of groups of premises disconnected from each other”.

Reply: The correction has been made.

Reviewer Point P 2.11 — Table 2 ”Assortativity degree” probably means ”degree assortativity” or ”Assortativity by degree”?

Reply: The correction has been made.

Reviewer Point P 2.12 — Table 2 Transitivity. This is defined for undirected networks only. In the directed case, there are more possibilities for triangular motifs. Explain.

Reply: To simplify, transitivity was calculated considering undirected networks. Triangle configurations were not discriminate. It has been developped in Table 2. Moreover, attempting to explain the networks structure and between farms contacts using mainly farms characteristics, we didn’t push the network analysis into such details.

Reviewer Point P 2.13 — Table 2 What does ”prop.nodes.Jaccar SC = 0” mean?

Reply: Thanks for noticing. It was initially the label used in Figures 3 and 4. It has been removed.

Reviewer Point P 2.14 — Table 2 Average distance. Be clear that the distance here is geo- graphical, as opposed to the network distances.

Reply: It has been clarified in the description.

Reviewer Point P 2.15 — l 158 08 kg − > 8 kg

Reply: The correction has been made.

Reviewer Point P 2.16 — l 165 ERGMs are not modified logistic regressions! They define an ensemble and new networks can be sampled from this ensemble. This is something different from a regression. Clarify.

Reply: The description was rephrased.

Reviewer Point P 2.17 — l 172 Typo: Model selection.

Reply: The correction has been made.

Reviewer Point P 2.18 — l 175 Use the correct letters for the names Erdos and Renyi. (The former are not available for this plain text review).

Reply: During the revision, the whole mention ’the simplest Bernoulli (“Erdo˝s–R´enyi”) model ’ seemed confusing between random Erd˝os-R´enyi model and random occurrence of edges (Bernoulli). We decided to simplify by only using ”the simplest model”.

Reviewer Point P 2.19 — l 188 MCMLE. Not explained. Also other (even though common) abbreviations, such as MCMC or MLE, should be written out when used for the first time.

Reply: Thank you for noticing, acronyms were clarified.

Reviewer Point P 2.20 — l 256 Typo: Analysis

Reply: Thanks for noticing, the correction has been made.

Reviewer Point P 2.21 — l 288 Assortativity can be computed for different properties. I guess you mean the degree assortativity here?

Reply: Yes, we refered to the degree assortativity described Table 2. It has been specified in the main text each time the term assortativity was used.

Reviewer Point P 2.22 — l 293 The longer..., the higher...

Reply: The correction has been made.

Reviewer Point P 2.23 — Table 5 Isn’t the ”sector of the industry” attribute redundant, when the ”type of premises” attribute is known? Please provide a brief explanation.

Reply: By using this “sector of the industry”, we attempted to gather multiple types of farms (breeders: MU & NU and producers: FA, PW, FI, FPW ) driven by the same factors to reduce the number of parameters in the selected models. It was detailed in table 5.

Reviewer Point P 2.24 — Table 5 Typos: 25th and 75th percentile.

Reply: Thanks for noticing, it has been corrected.

Reviewer Point P 2.25 — l 317 ”himself” -¿ ”the latter”?

Reply: Thanks for the suggestion, it has been replaced.

Reviewer Point P 2.26 — Table 6 Row 1: Does ’... for each movements appearing...’ mean ’for each movement pair’?

Reply: No, we meant for each movement created in the simulated network; e.g., for the distance, it would be the sum distances of each edge of the network. Thanks for pointing out the wording confusion. It has been clarified.

Reviewer Point P 2.27 — Table 6 Typo: e.g. (last row).

Reply: Thanks for noticing, it has been corrected.

Reviewer Point P 2.28 — l 323 Typo: ... all sub-networks.

Reply: Thanks for noticing, it has been corrected.

Reviewer Point P 2.29 — l 324 ’All network’ should be replaced by the whole network or so.

Reply: It was replaced by “all networks”.

Reviewer Point P 2.30 — l 324. typo: on average.

Reply: Thanks for noticing, it has been corrected.

Reviewer Point P 2.31 — l 337 growing pics models.

Reply: A unique model was selected for each type of animals. We think it should remain in the singular form.

Reviewer Point P 2.32 — l 334 Write out ’goodness of fit (Gof)’ at first appearance.

Reply: It was the written in the Material & methods part, line 196. We wrote it a second time in the result part, line 354.

Reviewer Point P 2.33 — Figures 5 and following Figures must be subdivided, i.e. Fig. 5 A, B, C, etc. in order to have clear references on the specific results. (And the figure axis contains ’premise’, see my general comment).

Reply: The word ’premise’ has been replaced by ’farm’, the Figures 5-7 were subdivided and the legends were adapted and rephrased.

Reviewer Point P 2.34 — Figures 5 and following The ordering of the figures is confusing. Consider either Gof followed by map or place 8, 9, 10 later.

Reply: Plos One guidelines imply to ”Place figure captions in the manuscript text in read order, immediately following the paragraph where the figure is first cited.”. Therefore, the order of figures 5 to 9 follows their order of appearance in the first paragraph which is more generic (L.353-364).

Reviewer Point P 2.35 — l 386. so -¿ though?

Reply: Thanks for the proposition, it has been replaced.

Reviewer Point P 2.36 — ll 417 ff ”nevertheless temporal network...” This is not related to the statement on clustering before. It should be related to the next sentence instead. Indeed, temporal and weighted networks are very interesting in this context and would give better results. However these sentences have no logical order and seem arbitrary. Please rephrase.

Reply: We wanted to emphasize the fact that although our static network-based approach can provide insight into pathogen flow patterns through a network, the use of temporal/dynamic analysis and contact chain analysis should be relevant tools for moving forward with this study. We restructured the sequence of sentences.

Reviewer Point P 2.37 — l 442. was detected.

Reply: Thanks for noticing, it has been corrected.

Reviewer Point P 2.38 — l 484. typo dependent.

Reply: Thanks for noticing, it has been corrected.

Reviewer Point P 2.39 — l 529 The term ’stochastic network’ means something else. Please rephrase.

Reply: “simulated” was use instead of the misplaced “stochastic” terms.

---

## [Decision Letter · Decision Letter 1]

16 Mar 2022

PONE-D-21-29344R1Complex network analysis to understand trading partnership in french swine productionPLOS ONE

Dear Dr. Hammami,

Thank you for submitting your manuscript to PLOS ONE. After careful consideration, we feel that it has merit but does not fully meet PLOS ONE’s publication criteria as it currently stands. Therefore, we invite you to submit a revised version of the manuscript that addresses the points raised during the review process.

The referees have provided a positive feedback and they consider the manuscript acceptable for publication. I am only asking to take into account the minor comments raised by the reviewers, that require minor changes and edits to the manuscript.  

We look forward to receiving your revised manuscript.

Kind regards,

Michele Tizzoni

Academic Editor

PLOS ONE

Journal Requirements:

Reviewers' comments:

Reviewer's Responses to Questions

**Comments to the Author**

1. If the authors have adequately addressed your comments raised in a previous round of review and you feel that this manuscript is now acceptable for publication, you may indicate that here to bypass the “Comments to the Author” section, enter your conflict of interest statement in the “Confidential to Editor” section, and submit your "Accept" recommendation.

Reviewer #1: All comments have been addressed

Reviewer #2: (No Response)

2. Is the manuscript technically sound, and do the data support the conclusions?

Reviewer #1: Yes

Reviewer #2: Yes

3. Has the statistical analysis been performed appropriately and rigorously? 

Reviewer #1: Yes

Reviewer #2: Yes

4. Have the authors made all data underlying the findings in their manuscript fully available?

Reviewer #1: No

Reviewer #2: Yes

5. Is the manuscript presented in an intelligible fashion and written in standard English?

Reviewer #1: Yes

Reviewer #2: Yes

6. Review Comments to the Author

Reviewer #1: Thank you for addressing the previous queries. These large-scale studies are difficult to bring together but the authors have done a good job at presenting the data.

Just a couple of clarifications:

L108 – Is the method in ref 8 for the animal introduction model described in the sentence after, starting ‘For each transportation…’ Currently it’s not clear. If it is described there, it would be more clear to put the reference after the explanation. E.g.

For each transportation, the records were divided according to the types of animals transported.

For each type of animal, directed movements were created from the loading farms to the

unloading farms when the loading event took place before the unloading events, as described in the method for the animal introduction model (8).

L208 – please can you clarify the length of a semester in the study?

There are still a few typological and grammatical errors in the text, I have suggested edits below:

French in the title should have a capital letter.

Abstract – ‘between-farm contacts’ , rather than 'between farms contacts'

Table 1 – production for sale or selling, rather than sell

Figure 1 caption: change amount to number (if that is correct). Quarantine typo, slaughterhouse typo. Suggest ‘by type of animal’ rather than ‘per type of animals’

L228 – ‘all holdings WERE involved’?

L240 – focusing on THE pig industry,

L243 – ‘varied ON average BY 1.7% per year’

L246 – ‘involved in between-farm movements’

L250 – intervalS

L251 – ‘unloaded onto a farm’

L253 – ‘whereas batches of 15…… and 2 …pigs were unloaded at slaughterhouses’

L268 - transportations or movements (plural)

L275 – aN association?

L278 – Model summaries

L279 – ‘as well aS model parameters’

Table 4 caption – ‘associated with’

L285 – mainly occurred

L287 – within THE breeding sector were in THE minority

L288 – did not change* this pattern

L288 – There were fewer, smaller weakly- and strongly-connected components in the breeding sector than in the growing sector.

L305 - 'The longer the studied time period, the higher the transitivity.'

L308- on average

L310 – on average

Fig3 – movements typo in figure in second facet on top row

L346 – involvements? What does this mean?

L353 suggestion - 'For all three models, the selected variables correctly reproduced the global

properties of the observed networks.'

L492 – in between-farm movements

Reviewer #2: The authors have addressed most of my comments. Thank you very much. Two minor points remain that should be addressed before publication of the manuscript.

ll 170 ff

"It aims, starting from a sample of a network...". The idea is to consider the observed network as one realization of all possible networks from the same ensemble.

"Samples can be used to infer the probabilities for a link...". This is not explanatory here (it rather explains link prediction). I am not sure whether the authors have understood the concept of ERGMs in full detail. However, please remove such sentences to avoid confusion. Rather restrict this part to more general explanations.

Table 2, Transitivity

Add a short note that link direction is ignored here.

7. PLOS authors have the option to publish the peer review history of their article (what does this mean?). If published, this will include your full peer review and any attached files.

Reviewer #1: No

Reviewer #2: No

---

## [Author Response · Author response to Decision Letter 1]

18 Mar 2022

Response to the reviewers

Once again, we wish to thank the editor and reviewers for their proofreading, suggestions, and corrections, which have helped us to improved the manuscript. In the following, we address reviewers comments.

Reviewer 1

Reviewer Point P 1.1 — L108 – Is the method in ref 8 for the animal introduction model described in the sentence after, starting ‘For each transportation…’ Currently it’s not clear. If it is described there, it would be more clear to put the reference after the explanation. E.g.

For each transportation, the records were divided according to the types of animals transported.

For each type of animal, directed movements were created from the loading farms to the

unloading farms when the loading event took place before the unloading events, as described in the method for the animal introduction model (8).

Reply: Thanks for the suggestion, the reference to the method has been moved to the end of the paragraph.

Reviewer Point P 1.2 — L208 – please can you clarify the length of a semester in the study?

Reply: The clarification was added.

Reviewer Point P 1.3 — There are still a few typological and grammatical errors in the text

Reply: Once again, we apologize for our English and sincerely thank the reviewer for his proofreading and suggestions. Correction have been made.

Reviewer 2

Reviewer Point P 2.1 — ll 170 ff

"It aims, starting from a sample of a network...". The idea is to consider the observed network as one realization of all possible networks from the same ensemble.

"Samples can be used to infer the probabilities for a link...". This is not explanatory here (it rather explains link prediction). I am not sure whether the authors have understood the concept of ERGMs in full detail. However, please remove such sentences to avoid confusion. Rather restrict this part to more general explanations.

Reply: Thanks for clarifying, both statements have been removed.

Reviewer Point P 2.2 — Table 2, Transitivity

Add a short note that link direction is ignored here.

Reply: The definition of transitivity in table 2 has been clarified.

---

## [Editor Report · Decision Letter 2]

22 Mar 2022

Complex network analysis to understand trading partnership in French swine production

PONE-D-21-29344R2

Dear Dr. Hammami,

We’re pleased to inform you that your manuscript has been judged scientifically suitable for publication and will be formally accepted for publication once it meets all outstanding technical requirements.

Kind regards,

Michele Tizzoni

Academic Editor

PLOS ONE
---

## [Editor Report · Acceptance letter]

29 Mar 2022

PONE-D-21-29344R2 

Complex network analysis to understand trading partnership in French swine production 

Dear Dr. Hammami:

I'm pleased to inform you that your manuscript has been deemed suitable for publication in PLOS ONE. Congratulations! Your manuscript is now with our production department. 

Kind regards, 

on behalf of

Dr. Michele Tizzoni 

Academic Editor

PLOS ONE